# Frequency-hopping wave engineering with metasurfaces

Hiroki Takeshita[1,4], Ashif Aminulloh Fathnan [1,2,4], Daisuke Nita[1,4], Atsuko Nagata[1], Shinya Sugiura [3] & Hiroki Wakatsuchi [1] ✉

Wave phenomena can be artificially engineered by scattering from metasurfaces, which aids in the design of radio-frequency and optical devices for wireless communication, sensing, imaging, wireless power transfer and bio/medical applications. Scattering responses vary with changing frequency; conversely, they remain unchanged at a constant frequency, which has been a long-standing limitation in the design of devices leveraging wave scattering phenomena. Here, we present metasurfaces that can scatter incident waves according to two variables—the frequency and pulse width—in multiple bands. Significantly, these scattering profiles are characterized by how the frequencies are used in different time windows due to transient circuits. In particular, by using more than one frequency with coupled transient circuits, we demonstrate variable scattering profiles in response to unique frequency sequences, which can break a conventional linear frequency concept and markedly increase the available frequency channels in accordance with a factorial number of frequencies used. Our proposed concept, which is analogous to frequency hopping in wireless communication, advances wave engineering in electromagnetics and related fields.

Materials available in nature essentially exhibit frequency dispersions or varying electromagnetic characteristics in accordance with the frequency spectra of incident waves.[1] Such materials are usually represented by linear time-invariant (LTI) systems,[2,3] which ensures that the same scattering profiles are obtained whenever a constant frequency is input. However, these time-invariant scattering profiles limit the degrees of freedom for controlling electromagnetic waves, and incoming signals containing particular frequency components. From an engineering perspective, the assignment of frequency resources is based on a *linear* frequency concept and maximized in accordance with the frequency bandwidth resolved.[4–6] In other words, the number of available frequency channels is proportional to the number of frequency slots used (Fig. 1a). However, the frequency channel number potentially increases if electromagnetic materials and their applied devices can behave differently within the same frequency resources. In particular, since controlling material response is limited

at a single frequency, achieving more degrees of freedom over more than one frequency will be important to break the conventional linear frequency concept and increase the frequency channel number and channel capacity[7].

Historically, artificially engineered subwavelength structures were intensively explored for more than half a century as frequency selective surfaces (FSSs),[8] metamaterials[9,10] and metasurfaces,[11] which successfully produced a wide breadth of applied devices and systems in the domains of, for instance, antenna design,[12] wireless communications,[13,14] sensing,[15] imaging,[16] wireless power transfer[17] and bio/medical applications.[18] However, as mentioned above, the electromagnetic response of these structures is also governed by frequency as they have frequency selectivity, hence limiting the degrees of freedom for electromagnetic wave manipulation. In recent years, efforts have been made to expand metasurfaces' capacity through additional parameters, thereby enabling multi-channel modulation

[1]Department of Engineering, Nagoya Institute of Technology, Gokiso-cho, Showa, Nagoya, Aichi 466-8555, Japan. [2]Research Center for Telecommunication (PRT), National Research and Innovation Agency (BRIN), Bandung 40135, Indonesia. [3]Institute of Industrial Science, The University of Tokyo, Meguro, Tokyo 153-8505, Japan. [4]These authors contributed equally: Hiroki Takeshita, Ashif Aminulloh Fathnan and Daisuke Nita. ✉e-mail: wakatsuchi.hiroki@nitech.ac.jp

even at the same single frequency. For example, a metasurface was reported to be capable of distinct wavefront manipulation in four circularly polarized output channels through the control of chirality-assisted phase response.[19] In another instance, polarization control has been utilized to introduce additional channels[20–22] for distinct holographic images associated with a right or left circular polarization, enhancing information encryption capacity in a photonic system.[20] Moreover, spatial variation of waveform-selective metasurfaces deployed around an omnidirectional antenna has enabled the selectivity of pulse width even under simultaneous incidence.[23] Despite these efforts to exploit all intrinsic local properties such as frequency, amplitude, phase, polarization and pulse width, the linear frequency concept remained the same and limited the available channel number as long as systems were constrained by LTI conditions.

A key solution to overcoming the limitations associated with LTI systems lies in introducing nonlinearity,[17,24–27] which provides capabilities superior to those obtained from a simple combination of linear media. For instance, the electromagnetic properties of metasurfaces no longer become time-invariant but rather are tunable if nonlinear media are used with electric or thermal stimulation[28,29] or optical pumping.[30] Alternatively, the use of nonlinear circuits[14,31,32] recently has led to the development of time-varying metasurfaces with advanced characteristics, including nonreciprocity,[33,34] harmonic generation[13,35] and spread spectrum capabilities.[36] These active reconfigurable metasurfaces have addressed unique challenges in various wireless communication scenarios such as space- and frequency-division multiplexing,[14] harmonic frequency modulation[37,38] and millimetre-wave wireless systems.[39] However, none of these studies have fully addressed the linear frequency concept issue or the limitation imposed by LTI conditions. Moreover, in practice, these advanced time-varying electromagnetic properties require precise (symbol-level) synchronization with transmitting sources as well as external control systems or energy resources such as direct-current (DC) sources, which limits the applicability of time-varying metasurfaces.

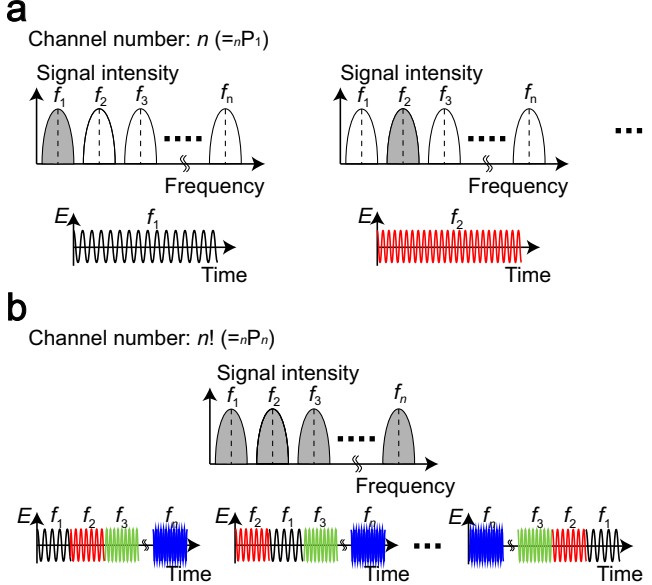

**Fig. 1 | Concept of the use of frequency-hopping metasurfaces to obtain additional degrees of freedom to engineer wave propagation.** (**a**) Conventionally available frequency channels. The frequencies used are limited to the frequency resolution realized. (**b**) Frequency channels extended by the proposed concept, specifically by frequency-hopping wave engineering. Even with the same frequency resources, signals are preferentially distinguished if their frequency sequences are different.

In this work, we introduce a new approach to increase electromagnetic wave selectivity or channel number using passive metasurfaces, which breaks the conventional linear frequency concept and overcomes the limitation imposed by classic LTI systems. Unlike previous metasurfaces that achieve variable scattering properties in accordance with frequency, polarization, amplitude, phase or pulse width of incident waves, we propose the use of multiple frequencies or, more specifically, *frequency sequences* as a new degree of freedom (Fig. 1b). Additionally, unlike active tunable metasurfaces that rely on external control systems, we develop passive yet variable metasurfaces that operate differently in response to the pulse width of incident waves but without any electrical biasing systems. Consequently, our metasurfaces have the potential to significantly expand the number of frequency channels while providing symbol-level synchronization, thanks to their passive and self-tunability operation. The proposed concept of obtaining distinct responses with variation in frequency sequences is akin to frequency hopping, the spread-spectrum modulation scheme used for Bluetooth,[40] and is utilized here as an analogue filter to spatially control electromagnetic waves.

## Results
### Theory
In this study, the pulse width is set to 50 ns or longer in the range of a few GHz, which ensures that the pulse spectrum is almost the same as the oscillating frequency.[41] However, under these circumstances, the scattering response essentially remains constant (see Supplementary Note 1). To overcome this constraint and achieve additional degrees of freedom, nonlinearity is an important factor since its use in electromagnetic media provides performance better than that from a simple combination of linear characteristics.[24] In particular, we use the recently proposed waveform-selective metasurfaces.[41–47] These nonlinear metasurfaces can be used to vary scattering parameters in accordance with the time width of an incident pulse (Supplementary Note 2) since the intrinsic resonant mechanisms of the metasurfaces are coupled to the transient responses of DC circuits. More specifically, we design a metasurface to selectively transmit an incoming wave by using the supercell depicted in Fig. 2a. Our metasurface is composed of a dielectric substrate (Rogers3003, 1.5-mm thick) and a conducting plane that has a periodic array of rectangular slits (7 mm × 15 mm) with a periodicity of 18 mm (see "Simulation model" in the Methods section for details). Lumped capacitors $C_1$ and $C_2$ are used to connect the gaps across the slits, which results in the adjustment of the resonant frequency. In addition, the gaps are connected by sets of four diodes that serve as diode bridge rectifiers. Thus, the incoming waveform is converted from a sine wave to its modulus waveform (i.e., |sin|) within the diode bridges, generating an infinite set of frequency components. However, most of the rectified energy appears at zero frequency, as seen from the Fourier series expansions (Supplementary Note 2). Because of the dominant zero-frequency component, transient responses appear when resistors are paired with inductors inside diode bridges.[41] Specifically, an inductor exhibits a strong electromotive force that blocks incoming electric charges during an initial period (Fig. 2b). An analogy is seen in a classic DC circuit comprising a DC power supply, a resistor and an inductor that shows a similar electromotive force during an initial period. Therefore, the lumped circuit elements are decoupled with the conducting geometry of the slit structure so that the intrinsic resonant mechanism is maintained to effectively transmit an incident wave. However, the electromotive force of the lumped inductors is gradually reduced by the zero frequency of the rectified electric charges (as seen in the above classic DC circuit), which results in the disruption of the resonance of the metasurface in the steady state.

We further exploit such waveform-selective mechanisms at several frequency bands by imposing $C_1 \neq C_2$ within metasurface unit cells. Importantly, waveform selectivity is related to DC circuit systems

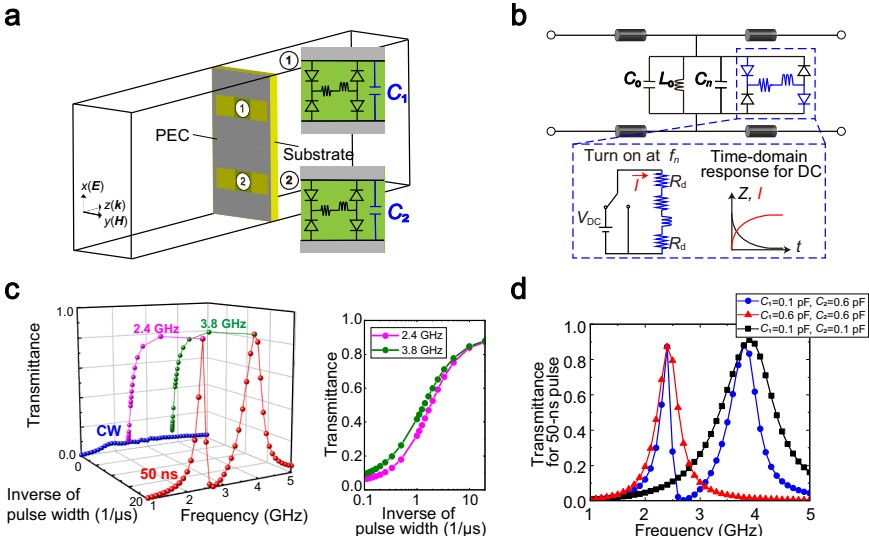

**Fig. 2 | Numerical demonstration of dual-band waveform-selective meta-surfaces.** (**a**) Supercell of the metasurface model based on the slit structure. $C_1$ and $C_2$ are used to adjust the operating frequencies of the unit cells. (**b**) Equivalent circuit model connected to the transmission line. The diode bridges across the slits and their internal circuit components (inside the top dashed box) can be approximated by a DC circuit, exhibiting a transient response even at the same frequency. (**c**) Scattering profiles (or transmittance profiles) extended from the classic frequency domain to the pulse-width domain. (**d**) Transmittance of 50-ns short pulses with $C_1 \neq C_2$ or $C_1 = C_2$. In these simulations, the input power is set to 10 dBm.

(Fig. 2b) activated by different frequency sources and therefore independent of each other. By either retaining the independence of the waveform-selective transient responses or introducing interlocking mechanisms, we demonstrate a particular type of wave scattering named frequency-hopping wave engineering.

## Multiband operation

First, we numerically designed and demonstrated a dual-band waveform-selective metasurface based on the slit structure shown in Fig. 2a (see "Simulation models" and "Simulation methods" in the Methods section). The simulated transmittance is plotted on the left of Fig. 2c, where $C_1$ and $C_2$ were set to 0.1 pF and 0.6 pF, respectively (see also "Definition of transmittances" in the Methods section). According to the simulation results, short pulses were more strongly transmitted than continuous waves (CWs) at 2.4 GHz and 3.8 GHz because of the aforementioned transient response. The right of Fig. 2c more clearly shows that the transmittance varied between -0.1 and 0.9 at the two different frequencies. These large transmittances at the two frequencies were attributed to the presence of the two different types of unit cells. Note that despite the limited physical area of the unit cells, the transmittance at these two frequencies was maximized due to the large effective area of the resonant cells covering the entire supercell. This observation aligned with findings from other reported works where efficient multiband operation arose from the combination of different subwavelength elements with distinct resonance frequencies.[48,49] Fig. 2d supports the observation that the transmittance peaks seen in the dual-band model were consistent with the transmittance peaks obtained by the single-band models, where $C_1$ and $C_2$ were both set to the same value of either 0.1 pF or 0.6 pF. Below, we show that our metasurfaces provide additional degrees of freedom for engineering electromagnetic wave propagation. Supplementary Note 3 provides more information on the simulation model of Fig. 2.

## Scattering based on frequency combination

The concept of waveform selectivity can be extended to more than two frequency bands by constructing supercells with additional unit cells with different capacitance values. Figure 3a shows a periodic supercell based on four unit cells using different capacitors. As seen in Fig. 2b, where a metasurface unit cell was related to a DC circuit, the supercell of Fig. 3a is associated with four independent DC circuits, each of which has a DC source activated by a different incident frequency. These circuit systems reduce the voltages across the inductors in the steady state, and thus, the structure in Fig. 3a is designed to show limited transient transmittance for a CW or a long duration waveform at any of the four resonant frequencies related to the unit cells (see also "Definition of transmittances" in the Methods section). However, if the incident frequency is regularly switched between these four frequencies, then the inductor voltages can possibly be restored to zero voltage or the initial condition to enhance transient transmittance again. Therefore, through the optimization of the transient response and recovery time, a large transient transmittance can potentially be obtained for a long waveform. This concept is schematically shown in the equivalent circuit model of Fig. 3b, where four DC circuits are equipped with either a DC source or one of the three short circuits, which are repeatedly swapped to enable the inductor voltage to recover to zero while other frequencies are in use (see $V_{L1}$ on the right of Fig. 3b). The numerical simulation results of the model in Fig. 3a are plotted in Fig. 3c and Fig. 3d for single-frequency source cases and switched-frequency source cases, respectively. In the former, the input frequency was fixed at 2.0, 2.5, 3.3 or 4.1 GHz. Note that Fig. 3c also shows the spectrogram for the single-frequency case using $f_1 = 2.0$ GHz. Under these circumstances, the transient transmittance was shown to be merely 0.05 or less since the circuit across the slit responding to the incident CW signal was short-circuited, which weakened the transmitting mechanism of the metasurface. However, the transient transmittance was enhanced by increasing the total number of frequencies available for switching at every 100 ns time step. A large transient transmittance was obtained despite the use of the same frequency components due to both the transient circuit response and the properly designed recovery time of the inductor circuits. As seen in Fig. 3d, with four frequencies available for switching ($f_1, f_2, f_3$ and $f_4$) the maximum transmittance reached around 0.4. Furthermore, this effect is more evident in Fig. 3e, where the entire pulse period (i.e., one set of four pulse durations) was equally distributed among the four frequencies and varied from 0.1 μs to 30 μs. According to this simulation result, the transmittance averaged over the entire pulse period was maximized at a few hundred nanoseconds and then gradually decreased with increasing pulse period. This phenomenon occurred as

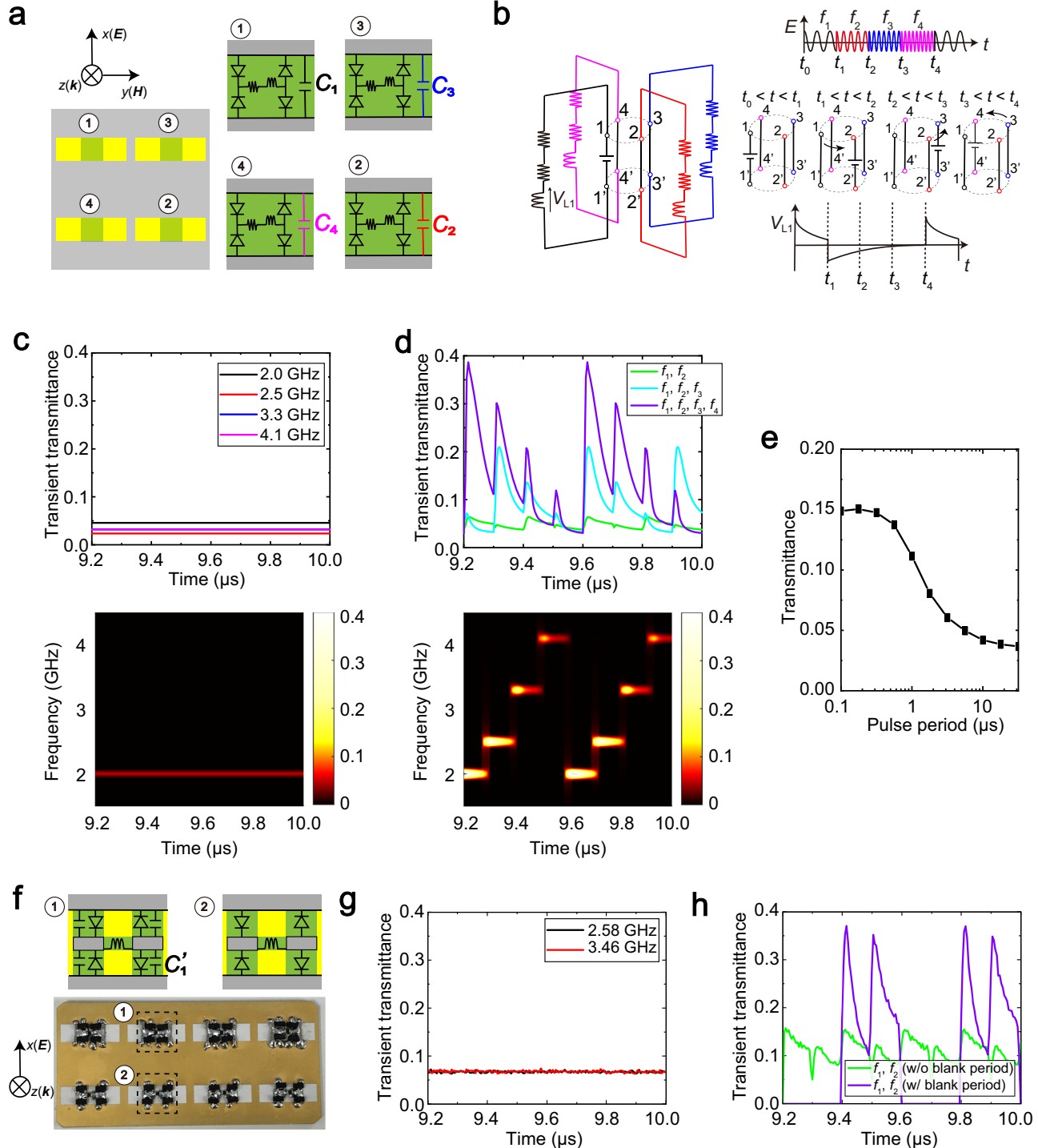

**Fig. 3 | Numerical and experimental demonstration of engineering wave propagation in accordance with frequency combinations.** (**a**) Supercell of quad-band waveform-selective metasurfaces. $C_1$, $C_2$, $C_3$ and $C_4$ are 0.1, 0.3, 0.6 and 1.1 pF, respectively. (**b**) Equivalent circuit system (left) and expected time-domain profiles (right). In the time domain, the incident frequency is repeatedly changed (top right), which corresponds to changing the position of the DC source (middle right). An inductor voltage can be restored to zero voltage while other frequencies are used (bottom right). (**c**) Simulated transient transmittance for single-frequency cases (top) and spectrogram of the transient transmittance for 2.0 GHz (bottom).

(**d**) Simulated transient transmittance for switched-frequency cases (top) and spectrogram using all of the four frequencies (bottom). $f_1$, $f_2$, $f_3$ and $f_4$ are 2.0, 2.5, 3.3 and 4.1 GHz, respectively. (**e**) Average transmittance as a function of the entire pulse period. (**f**) Measurement sample designed to operate at two frequencies within a standard rectangular waveguide. (**g**) Measured transient transmittance for single-frequency cases. (**h**) Measured transient transmittance for the switched-frequency case. $f_1$ and $f_2$ are adjusted to 2.58 and 3.46 GHz, respectively. In these simulations and measurements, the input power is set to 10 dBm.

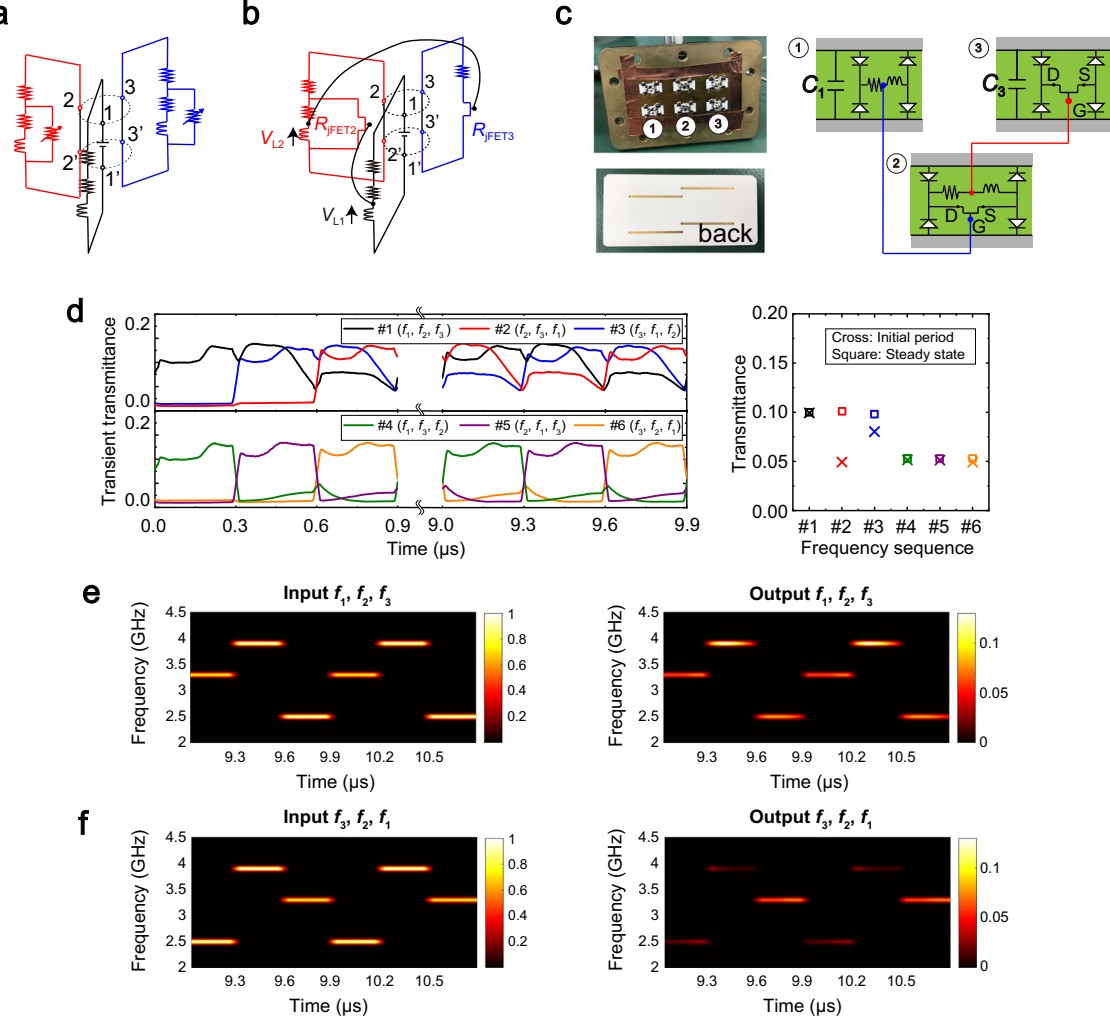

**Fig. 4 | Experimental demonstration of engineering wave propagation in accordance with frequency sequence.** (**a**) Equivalent circuit system with variable resistors. The variable resistance values change in the time domain depending on the incoming frequency sequence. In the time domain, the incident frequency is repeatedly changed in a particular sequence, which determines how the position of the DC source is moved. (**b**) Specific circuit system. JFETs are included and biased by other circuits. (**c**) Measurement sample designed to operate at three frequencies. (**d**) Measured transient transmittances during the initial period and in the steady state (left) and their averages (right). $f_1$, $f_2$ and $f_3$ are 3.3, 3.9 and 2.5 GHz,

respectively. During the initial period, the number of distinct frequency sequences is determined by $N!$, where $N$ represents the number of frequency channels available. In contrast, in the steady state, the number of distinct frequency sequences is reduced to a circular permutation of $N$, namely, $(N-1)!$. (**e**) Normalized spectrogram of the input and output signals (left and right, respectively) using the frequency sequence of $f_1$, $f_2$ and $f_3$. (**f**) Normalized spectrogram of the input and output signals (left and right, respectively) using the frequency sequence of $f_3$, $f_2$ and $f_1$. (**e**) and (**f**) correspond to sequences #1 and #6 in (**d**), respectively. In these measurements, the input power is set to 20 dBm.

the transmittance approached the steady-state condition, requiring an extended duration for the restoration of the inductor voltage. Therefore, a moderately long pulse period is important for efficiently increasing the transmittance. The use of the proposed concept provides not only waveform selectivity at several frequency bands but also an additional parameter for designing electromagnetic scattering, even at the same frequencies, which surpasses the capabilities of conventional frequency selectivity or two-parameter schemes. Thus, our design provides additional degrees of freedom for engineering wave propagation.

We experimentally validated such a waveform-selective metasurface operating in multiple frequency bands using the measurement sample shown in Fig. 3f (see "Measurement samples" and "Measurement methods" in the Methods section). This sample comprised four sets of two different unit cells with/without additional capacitors to vary the operating frequency and was designed to operate at 2.58 GHz and 3.46 GHz. The measured transient transmittance for a single-frequency CW oscillating at only one of these frequencies was ~0.07

(Fig. 3g). However, with the switching of the oscillation frequency between these two frequencies, the transient transmittance increased, as shown in Fig. 3h. Importantly, owing to the limitation of the bandwidth of the rectangular waveguide used (see "Measurement methods" in the Methods section), this structure was designed to operate at only two frequencies, unlike the simulation model, which was designed for four frequencies. Therefore, the improvement in the measured transient transmittance was relatively limited compared with that in the simulated transient transmittance (green curve in Fig. 3h and blue curve in the top of Fig. 3d). However, a comparable level of transient transmittance was observed when an additional 200 ns was used for the entire pulse period, which ensured that the pulse period in the measurement was the same as that in the simulation (blue curve in Fig. 3h). We expect the measured transient transmittance to further improve with an increasing number of unit cells or operating frequencies. These results support the theory that waveform selectivity is feasible, and more importantly, the transient transmittance increases with the use of waveform selectivity or the transient circuit response at

several frequencies. Additional numerical and experimental results related to Fig. 3 are shown in Supplementary Note 4 and Supplementary Note 5, respectively. Also, Supplementary Note 6 reports how the transmitting performance is further improved.

## Scattering based on frequency sequence

The scattering concept extended by frequency combination depicted in Fig. 3 can demonstrate even more selectivity for engineering incoming signals containing the same frequencies if the scattering is determined by a particular sequence of frequencies. Such selective scattering may be achieved using the schematic shown in Fig. 4a. This circuit configuration is similar to the one shown in Fig. 3b (despite the minor difference in the number of DC circuit systems). Additional variable resistors are introduced to maintain or disrupt each transient response. For instance, if the variable resistors have a large resistance, then a transient circuit response is obtained, as seen earlier. However, if these resistances are small, the transient response disappears, signifying that the transmittance of the metasurface is lowered. The resistances are controlled by coordinating each circuit system or the corresponding frequency with one of the other frequencies, which leads to scattering control based on the frequency sequence. This concept is achieved using a junction-gate field-effect transistor (JFET), as shown in Fig. 4b. The inductor voltage $V_{L1}$ is applied to the gate of the JFET in the left circuit as the bias voltage. Depending on the inductor voltage (with or without the incident wave to activate the left DC circuit), the effective resistance $R_{JFET2}$ between the drain and source changes, which varies how the left circuit behaves when the DC source is moved to the left circuit (i.e., when the incident frequency component is changed). Similarly, the inductor voltage $V_{L2}$ of the left circuit is used to switch the state of the right circuit. Since such a structure involves complicated nonlinear phenomena and causes instability during simulation, we experimentally validated our design using the measurement sample shown in Fig. 4c (see "Measurement samples" and "Measurement methods" in the Methods section). We combined three 300-ns pulses, for which the oscillation frequencies were 2.5, 3.3 and 3.9 GHz. When the input frequency was changed in the sequence of 3.3, 3.9 and 2.5 GHz, the entire transient transmittance was enhanced most strongly, as seen in Fig. 4d (see frequency sequence #1). With other frequency sequences, some of the JFETs were not properly activated, which resulted in lower transient transmittance. Such selectivity based on the frequency sequence can be used for longer waveforms, as demonstrated in Fig. 4d. The transmittance gap in the steady state is more evident in the spectrograms shown in Fig. 4e and Fig. 4f. In these results, the output signal exhibits a consistent level of transmittance across all three frequencies if the incident signal is excited at frequency sequence #1 in Fig. 4d instead of frequency sequence #6. Note that the steady-state case had only two distinguished sequences since only three frequencies were used. Consequently, sequences #1, #2 and #3 demonstrated consistent high-level transmittance, while sequences #4, #5 and #6 displayed corresponding low-level transmittance. By increasing the number of frequencies used $N$, we can attain more degrees of freedom, specifically in accordance with $N!$ (i.e., the factorial of $N$) during the initial period and $(N-1)!$ in the steady state. These results support the theory that electromagnetic scattering or wave propagation can be engineered according to the sequence of frequencies, which cannot be realized by the conventional concept of frequency selectivity or existing LTI systems. Supplementary Note 7 summarizes more results related to Fig. 4 as well as design strategies for preferable sequence of frequencies within the same supercell configuration.

## Potential application in wireless communications

As an example of potential applications, the concept of our metasurfaces is leveraged in wireless communications where additional selectivities are made available in accordance with particular frequency sequences. Metasurfaces function as spatial filters, similar to the well-known concept of FSSs that preferentially transmit a signal depending on the incident frequency. In contrast to classic FSSs, here, our metasurface shown in Fig. 4 selectively transmits signals in accordance with the frequency sequence. To evaluate such a scenario, we conducted experiments on our metasurface involving the transmission of realistic communication signals containing binary data. Specifically, we examined the relationship between the bit error rate (BER) and the signal-to-noise ratio (S/N) when utilizing the metasurface for communication with binary phase shift keying (BPSK) modulation and frequency-hopping carriers. The measurement setup for this evaluation is depicted in Fig. 5a, where the incident signal was subjected to BPSK modulation (see "Modulation method" in the Methods section in detail). After passing through the metasurface, the signal was demodulated in the presence of additive white Gaussian noise (AWGN). It is important to note that the diagram blocks representing the transmitter and receiver in Fig. 5a are adopted from the spread spectrum technique widely known as frequency-hopping spread spectrum (FHSS).[40] The left panel of Fig. 5b represents the received BPSK signals without AWGN, while the right panel shows the received signals with AWGN. BER calculations at the receiver's end indicate that when the metasurface was utilized with the correct frequency sequence ($f_1$, $f_2$ and $f_3$ or sequence #1 in Fig. 4d), the BER remained low, indicating successful transmission. However, when the frequency sequence differed from the predetermined sequence (specifically, $f_3$, $f_2$ and $f_1$ or sequence #6 in Fig. 4d), the BER increased, indicating errors in data transmission. The difference between the BERs became more pronounced when the S/N increased, as depicted in Fig. 5c. For example, at S/N = 5 dB, the difference reached almost tenfold. Supplementary Note 8 provides further insights into how this difference in BER impacts the transmission of realistic image data.

Importantly, Fig. 5c demonstrated that BER performance is improved in the case of the preferred frequency sequence ($f_1$, $f_2$ and $f_3$ or sequence #1), which clearly indicates that the metasurface works as a spatial filter. This novel functionality opens the door to numerous potential applications in wireless communications. For example, the concept of our metasurfaces can be utilized to realize secure electromagnetic buildings or smart shielding.[50,51] In this application, by placing a metasurface as a selective window for communication, signals are allowed to enter a building only if the frequency-hopping sequence is correct, while other signals are rejected even with the same frequency resources. Thus, our concept can contribute to confining confidential radio signal communications in designated areas of interest, enhancing wireless security. Furthermore, the concept of frequency-hopping selectivity can be used not only as spatial filters but also as smart antennas. Such an antenna design holds promising potential across various domains, including physically unclonable functions (PUFs)[52–54] and internet of things (IoT) tags.[55,56] In the context of IoT tags, the additional degree of freedom in accordance with the frequency sequence provides more identities (IDs) and thus permits the simultaneous use of more IoT tags within the same wireless network.

Furthermore, wireless communications utilize wider bandwidths in realistic modulation schemes,[40] as opposed to Fig. 2 to Fig. 4, where incident waves were excited at single frequencies.[40] Therefore, our metasurface demonstrated in Fig. 5d to Fig. 5f was experimentally evaluated by using chirp signals that swept the oscillation frequency during each 300-ns time slot as shown in Fig. 5e (Fig. 5d). In this measurement, we used frequency sequences #1 and #6 shown in Fig. 4d. Under this circumstance, the measurement results plotted in Fig. 5f demonstrate that selectivity based on frequency sequences still appeared even if the bandwidth of the incident signal was broadened. For instance, when the oscillation frequency was changed between ± 200 MHz (i.e., the bandwidth wider than that used for Bluetooth and Wi-Fi[40]), the transmittance

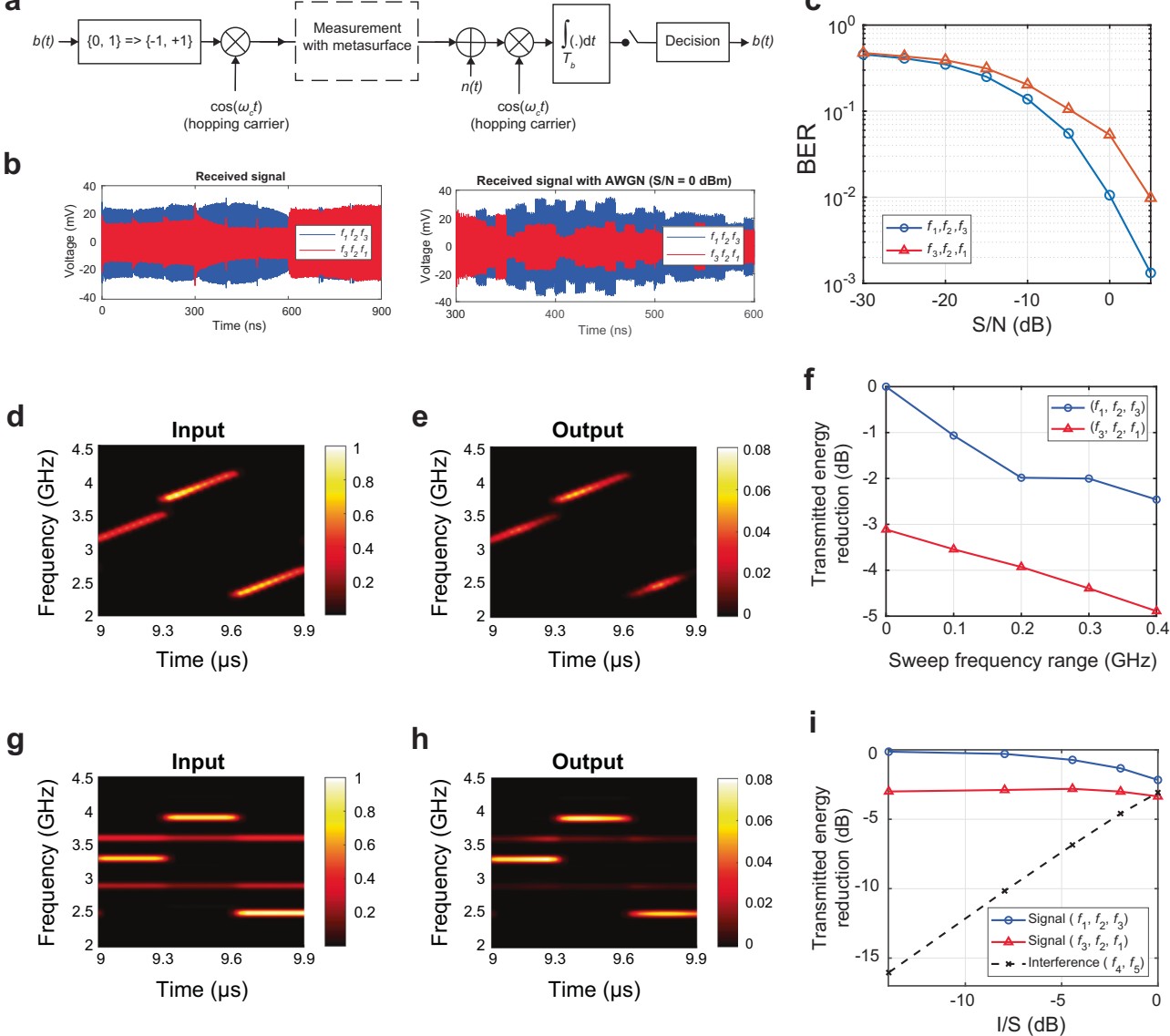

**Fig. 5 | Experimental demonstration of the metasurface towards realistic communication scenarios.** (**a**) Diagram block of communication scenario using the proposed metasurface and the BPSK modulation scheme. The time length for each bit $T_b$ is set to $T_b = 10$ ns, while the time slot for each frequency carrier is 300 ns (corresponding to 30 bits). (**b**) Examples of the received BPSK signals comparing two frequency sequences (left) before and (right) after implementing additive white Gaussian noise. (**c**) BER vs. S/N analysis using two different frequency sequences. (**d**, **e**) Spectrogram of (**d**) input and (**e**) output chirp signals sweeping the oscillation frequency during each 300-ns time slot. The centre frequency in each time slot hops between $f_1$, $f_2$ and $f_3$, which correspond to sequence #1 in Fig. 4d. In this example, the chirp signal is swept by 0.4 GHz in each time slot. (**f**)

Transmittance reduction with different sweep frequency ranges. The transmitted energies of chirp signals using different frequency sequences are compared to the transmitted energy of sequence #1 shown in Fig. 4d. (**g**, **h**) Spectrogram of (**g**) input and (**h**) output frequency-sequence signals corresponding to sequence #1 in Fig. 4d with double-band interference signals at $f_4 = 2.9$ GHz and $f_5 = 3.6$ GHz. (**i**) Transmittance reduction with different interference-to-signal ratios (I/Ss). The transmitted energies of frequency-sequence signals and interference signals are compared to the transmitted energy of sequence #1 shown in Fig. 4d. Here, the interference signal magnitude ($f_4$ and $f_5$) is swept following the interference-to-signal ratio (I/S) from −14 dB to 0 dB. In these measurements, the input power is set to 20 dBm.

difference between the two frequency sequences was maintained at 3 dB.

In addition, we conducted measurements on our metasurface using interference signals. Notably, in practical wireless communication scenarios, multiple frequency bands are often utilized simultaneously to enhance data transmission efficiency.[57] However, the concurrent use of neighbouring frequency bands can lead to significant interference with intended communication signals.[40] As depicted in Fig. 5g and Fig. 5h, we intentionally employed neighbouring frequencies ($f_4 = 2.9$ GHz and $f_5 = 3.6$ GHz) as interference signals. The interference-to-signal ratio (I/S) was then varied by changing the magnitude of the interference signals to observe the impact on the

transient transmittance of the original signal. Remarkably, Fig. 5i demonstrates the metasurface's robustness against interference from neighbouring frequencies, as the interference signals were effectively suppressed while maintaining the transmittance of the original signal (see Supplementary Note 8 for detailed results). This resilience can be attributed to the metasurface's inherent characteristic of selectively allowing transmittance for specific intended narrow bands. While the narrow-band characteristics are advantageous in suppressing interference and improving communication performance, this holds true only when the signal bandwidth is narrower than the metasurface's bandwidth. In cases where the signal bandwidth exceeds the metasurface's bandwidth, design adjustments must be made to avoid

deterioration of the overall communication performance. Consequently, the results depicted in Fig. 5 support the potential application of our metasurfaces in wireless communications, offering additional selectivity based on frequency sequences.

## Discussion

The proposed concept offers capabilities beyond those of conventional frequency selectivity approaches. In the literature, wave scattering has been controlled by both frequency and pulse width, which adds an additional degree of freedom to classic FSSs or metasurfaces.[41,44,58] In this study, we extended the waveform selectivity to more than one frequency band through multiband structures. Such multiband operation is observed in various fields, including metasurfaces and their applications.[1,8,59,60] However, unlike prior studies, our design provides more freedom to characterize electromagnetic properties/responses as well as their applied devices by properly designing the transient response of loaded circuits, their recovery time and coupling, which yields an additional selectivity based on the factorial number of frequencies available. In particular, our metasurfaces can be utilized for advanced spatial filters to design wireless communication environments and to more efficiently use limited frequency resources even at the same frequencies.[4–6,40] While several relevant efforts on linear and nonlinear metasurfaces have reported advanced wave selectivities, such as polarization dependency,[20–22] angular memory[61] and pulse-width dependency,[23,44,47,58] none of the existing studies have exploited selectivity based on frequency sequences. In particular, multifunctional metasurfaces with broadband absorption and waveform selectivity for free space waves have been reported but without frequency-sequence dependency.[46] Wideband FSSs have also been designed to distinguish different pulsed waveforms.[62,63] However, the intrinsic mechanism was linear and had no time variation. Also, time-varying metasurfaces have been utilized to increase the number of carrier frequencies through harmonics generation, while the number of channels was still linearly proportional to the frequencies generated by signal sources and modulated metasurfaces.[64,65] Therefore, the linear frequency concept issue remained unchanged so far but was successfully overcome by our metasurfaces. Moreover, although our concept appears analogous to the principle of the frequency-hopping spread spectrum (FHSS), it is also conceivable to design metasurfaces with a phase-hopping technique to conform to the direct sequence spread spectrum (DSSS). In such a scenario, the metasurfaces would incorporate phase modifications while maintaining a consistent carrier frequency within a specific frequency band. This approach may offer enhanced immunity to narrow-band interference and facilitate simpler synchronization techniques, which are based on only single frequency operation. However, the metasurfaces need to be sensitive to phase variations to accurately replicate the pseudorandom phase coding in the transmitted signal, necessitating further investigation in future studies.

As an example of potential applications of our metasurfaces, we presented measurement results in Fig. 5 to show how our metasurfaces can be leveraged in wireless communications. Importantly, while our evaluation focused on one-dimensional systems for simplicity, metasurfaces are designed over two-dimensional surfaces in more practical wireless communication scenarios. This expanded design space offers increased degrees of freedom, enabling additional capabilities such as optical angular momentum and angular dependence.[61,66] In particular, as extensively explored in recent years, our metasurfaces can be utilized for beamforming when the metasurfaces exhibit a phase gradient pattern over two-dimensional surfaces to achieve anomalous reflection/transmission.[67] Additionally, in Supplementary Note 4 we showed that our metasurfaces are not limited to one-dimensional systems but can be extended to respond to free-space waves as a first step towards applications in realistic wireless communication environments.

Despite the limited transmittance observed in the current metasurface design illustrated in Fig. 3e and Fig. 4d, we emphasize the availability of diverse strategies amendable to our metasurfaces for enhancing their transmittance, including optimizing the resonance's $Q$ factors. Supplementary Note 6 provides information on how the transmitting characteristics can be further improved by reducing the resonance's $Q$ factors, which helps our metasurfaces more efficiently work in specific application scenarios. To validate this approach, we specifically design such metasurfaces and present measurement results of transmittance that is markedly improved, compared to the slit structure design used above (see Supplementary Note 6 for details). It is worth noting that our effort to enhance transmittance by lowering the $Q$ factor via manipulating the meta-atom configuration represents just one instance of such attempts. Potentially, other approaches are applicable to further improve transmittance, for instance, by changing material properties or by utilizing multi-layer metallic configurations. Importantly, we also note that the transmittance limitation primarily arises from parasitic elements inherent in the embedded circuits, notably the diodes. Given the current configuration, the inclusion of these lossy diodes is imperative. Hence, prospective investigations could focus on developing unit cells equipped with specially designed low-loss diodes, potentially with an optimization from the semiconductor device level.

In recent years, metasurfaces have gained significant attention for sixth-generation (6 G) mobile communication systems, particularly as simplified analogue repeaters for millimetre-wave communications.[68–70] This surge of interest in practical applications is a relatively recent development, although the extensive studies on metasurfaces have been conducted over the past two decades. Consistent with this trend, our proposed concept of analogue spatial frequency-hopping (frequency-sequence) filters offers the potential to transform the design approaches in future communication systems. One promising application of our metasurfaces is in physical-level information encryption, where they can be utilized within secure communication systems that leverage beamforming techniques. As described by the wiretap channel model,[71–73] our metasurfaces can increase the secrecy capacity by maximizing the beamforming gain towards the legitimate receiver while intentionally degrading the signal quality at the eavesdropper's location. Although our current proposed scheme does not directly create a beam or null for physical layer security purposes, further research can be conducted to realize beamforming techniques specifically tailored to the selectivity based on frequency sequences in our metasurfaces. Potential directions for future work could involve leveraging the periodicity of the proposed metasurfaces within the supercells to engineer diffraction into various angles, thereby enabling controlled beamforming. Another approach could involve utilizing a multilayer structure, with one layer acting as a beamformer while our metasurfaces remain in the other layer. Note that similar works have realized beamforming metasurfaces based on the incoming pulse width for cloaked antennas[23] and reconfigurable intelligent surfaces (RISs).[67]

We proposed a scattering paradigm that is not limited by the conventional concept of frequencies or characteristics of LTI systems. We theoretically presented a methodology to design metasurfaces loaded with circuit components such as diodes and numerically and experimentally validated the ability of these metasurfaces to distinguish different waves even at the same frequencies in accordance with their pulse widths. Our metasurfaces were designed to operate in more than one frequency band, which signifies that the scattering profile can be designed fully by using frequency and pulse width. Importantly, the metasurfaces enhanced transient transmittance if the incident frequency was changed (or hopped) repeatedly in particular combinations or sequences, which has successfully overcome the issue related to the conventional linear frequency concept. Thus, our study introduces a new approach for designing electromagnetic materials[1,10] and

related devices and systems[12,59] in wireless communications,[13,14] sensing,[15] imaging,[16] wireless power transfer[17] and bio/medical applications,[18] which are no longer bound by the number of frequencies but rather can be extended to a larger framework based on frequency-hopping wave engineering.

## Methods

### Simulation models

Our simulation models were based on slit structures. Evenly spaced rectangular apertures were constructed in the conducting plate (perfect electric conductor: PEC) on a dielectric substrate (1.5-mm-thick Rogers3003) with periodic boundaries. In the default circuit configuration or an inductor-based circuit, each aperture was bridged by a set of four diodes (Broadcom HSMS286x series) where a resistor was connected to an inductor, as shown in Fig. 2 to Fig. 4. In Fig. 4, JFETs (Toshiba 2SK880-BL) were used to maintain or disrupt the transient response of the inductor circuit. The specific design dimensions and parameters are given in Supplementary Notes 3 to 7.

### Simulation method

We adopted a cosimulation method[41–43] to facilitate the design process of metasurfaces and optimized their performance by using the ANSYS Electronics Desktop Simulator (2020 R2). In this method, an electromagnetic metasurface model was first simulated in an electromagnetic simulator (HFSS). Lumped circuit components were replaced with lumped ports. These electromagnetic scattering profiles were used in a circuit simulator where lumped ports were connected to actual circuit components such as diodes, resistors and inductors. This was effectively the same as directly connecting the circuit components to the electromagnetic model via the lumped ports in electromagnetic simulations. However, the use of the cosimulation method markedly improved the simulation efficiency.

### Definition of transmittances

In this study we evaluated transmission characteristics using two terminologies, "(ordinary) transmittance" and "transient transmittance". The former transmittance was used in most simulations and measurements (e.g., Fig. 2c and Fig. 2d) to assess the frequency- and time-domain transmittances by dividing the total transmitted energy by the total incident energy. In contrast, the latter transmittance (i.e., transient transmittance) was used to more properly evaluate time-varying transmission characteristics in the time domain (e.g., in the top of Fig. 3c and Fig. 3d) by dividing the moving average of the transmitted energy by that of the incident energy. However, when the entire performance of the time-varying characteristics was calculated (e.g., in Fig. 3e and the right panel of Fig. 4d), "(ordinary) transmittance" was still used.

### Measurement samples

Measurement samples were designed based on the corresponding simulation models (see Supplementary Notes 5 to 7 for the dimensions and parameters used for the measurement samples). However, owing to the spatial limitations of the measurement setups, the number of unit cells was limited to six or eight. Circuit components were soldered to the conducting surfaces of the measurement samples.

### Measurement methods

To evaluate the experimental performance of the measurement samples, frequency-domain characteristics were evaluated using a vector network analyser (VNA) (Keysight Technologies N5249A). The VNA was connected to coaxial cables and then a standard rectangular waveguide (WR284), where the measurement samples were fixed using copper tape. Time-domain results were obtained using an arbitrary waveform generator (AWG) (Keysight Technologies M8195A) and an oscilloscope (Keysight Technologies DSOX6002A).

### Modulation methods

In addition to the above time-domain measurement methods, we evaluated the communication performance of the proposed metasurface. Here we used a system model involving the BPSK modulation scheme with a carrier frequency that hopped every 300 ns in the sequence of $f_1 = 3.3$ GHz, $f_2 = 3.9$ GHz and $f_3 = 2.5$ GHz or in the reversed sequence. A text file of the BPSK-modulated signal was first generated using MATLAB and input into the abovementioned AWG. The signal was sent to the rectangular waveguide (WR284) in which the metasurface was placed. AWGN was added to the transmitted signal to model a realistic noisy channel and conduct BER vs. S/N analysis. Figure 5a shows the block diagram of the system model where $b(t)$ denotes the input binary data and $\omega_c$ is the angular frequency of the carrier signal (hopping between $f_1$, $f_2$ and $f_3$ with two distinct sequences). In the demodulation process, the received signal was multiplied by the same carrier frequency. An integrator and a threshold detector were used to remove harmonics and retrieve the corresponding bits, respectively.

## Data availability

The data that support the findings of this study are available from the corresponding author upon request.

## Code availability

The codes that are used to generate results in the paper are available from the corresponding author upon request.

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

## Acknowledgements

This work was supported in part by the Japan Science and Technology Agency (JST) under Precursory Research for Embryonic Science and Technology (PRESTO) No. JPMJPR1933 and JPMJPR193A and under Fusion Oriented Research for Disruptive Science and Technology (FOREST), Japan Society for the Promotion of Science (JSPS) KAKENHI No. 17KK0114 and No. 21H01324 and the Japanese Ministry of Internal Affairs and Communications (MIC) under Strategic Information and Communications R&D Promotion Program (SCOPE) No. 192106007.

## Author contributions

H.W. conceived of the idea and designed the project. H.T., A.A.F. and D.N. primarily performed simulations and measurements under the supervision of H.W. and S.S. A.N. also supported the measurements. All authors contributed to analyzing the results and editing the paper.

## Competing interests

The authors declare no competing interests.
