## [Peer Review File · Nature Communications]

Reviewers' Comments:

Reviewer #1:

Remarks to the Author:

Metasurfaces that scatter incident waves differently with various frequency and pulse width in multiple bands, are examined. Scattering profiles are characterized by how frequencies are used in different time slots due to transient circuits. The circuits are comprising capacitors, diodes and switches while being coupled with slits constituting the metasurface supercell. In time domain, the change in the incident frequency corresponds to changing the position of the DC source which initializes and flips the sign of the response. The average transmittance as a function of pulse period is represented while the response when the frequency is switched, can be dramatically controlled from the duration of the pulse. Measurements somehow validate the simulation result.

The paper analyzes an interesting idea of the well-known frequency hopping translated into photonic communication with metasurfaces. However, it is not suitable for a publication at Nature Communications in its present form. In particular:

(A) The assessment of the quality of the communications should occupy a significant part of the paper. For example, a typical drawback of frequency hopping is the interference between the transmissive waves of neighboring frequencies. The authors should accompany their results with an evaluation of such a figure of merit.

(B) The authors should examine the possibility of expanding their paradigm to phase hopping and provide the reader with a list of pros and cons against their current frequency hopping.

(C) The authors may discuss the connection of their concept with other metasurface-based communications setups involving secure transmission [1,2].

(D) The whole structure is one-dimensional. However, fields diffuse across all directions. How the analysis is modified if two-dimensional waves are utilized as happening in other memory structures [3,4]?

[1] Metasurface-Coated Devices: A New Paradigm for Energy-Efficient and Secure 6G Communications, IEEE Vehicular Technology Magazine, 2021.

[2] Metasurface-Assisted Wireless Communication with Physical Level Information Encryption, Advanced Science, 2022.

[3] Reconfiguring structured light beams using nonlinear metasurfaces, Optics Express, 2018.

[4] Angular memory of photonic metasurfaces, IEEE Transactions on Antennas and Propagation, 2021.

Reviewer #2:

Remarks to the Author:

In the manuscript "Frequency-Hopping Wave Engineering with Metasurfaces" the authors utilized diode bridge rectifiers in the meta-atoms to enable designed transient responses, which further improve the transmittance of EM waves. However, this phenomena only exist for a short time, let alone the transmittance is even dramatically varying. All these drawbacks make it hard to be used in modern electronic systems. Therefore, unless the authors clarify the application scenarios of such kind of device more clearly, I could not recommend it be published in Nature Communications. In addition, there are some other suggestions.

1. The definition of transmittance should be provided clearly. For example, in Figs. 3d, h and 4d, does it refer to the transmittance of the transient frequency?

2. In Fig. 2c, the descriptions of the blue and red lines are missing.

Reviewer #3:

Remarks to the Author:

The paper proposes a kind of metasurface that can manipulate scattering waves according to the incidence frequency and pulse width in the designed bands. The metasurface loads different coupled transient circuits for different frequency and waveform selection, showing the scattering based on frequency combination and sequence. Numerical and experimental results demonstrate its efficiency of frequency-hopping wave engineering.

My comments and suggestions are as follows:

1. Some relative electromagnetic engineering works have been done, the new contribution over the work from the same group [1] is incremental. The design of rectifier-based circuit is a common technique in the antenna and propagation society. Much more comprehensive comparisons with the state of arts should be supplemented to highlight the novelty. Moreover, the measured transmission is very low, it is difficult to be employed in realistic applications.

[1] D. Ushikoshi, R. Higashiura, K. Tachi, A. A. Fathnan, S. Mahmood, H. Takeshita, and H. Wakatsuchi, "Pulse-driven self-reconfigurable meta-antennas," *Nature Communications*, 14(1), 633, 2023.

2. Fig. 3 (d), (h), and Fig. 4(d) show that the transmittance of designed supercells all less than 0.4, performing low efficiency. And some cases (for example incident switched-frequency is f_1 and f_2 in fig. 3 (d)) are even worse. The authors should have an explanation.

3. In the section 2, it is not very clear to me how the waveform selection is working, although there are some references are mentioned 29-33. The authors should explain its mechanism with more details.

4. In Fig. 4d, the measured average transmittance during the initial time period and in the steady state are not consistent in the cases of frequency sequence #2 and #3, while in other cases of frequency sequences are the same. The authors should have an explanation.

5. The paper suffers from bad English expression in general, and it makes me confused when I read the paper.

6. Line 129, "Fig. 2b" should be "Fig. 3b", please the authors confirm it.

7. Line 372, "(g)" should be "(d)", please the authors confirm it.

Responses to Reviewer 1

Comment 0: *Metasurfaces that scatter incident waves differently with various frequency and pulse width in multiple bands, are examined. Scattering profiles are characterized by how frequencies are used in different time slots due to transient circuits. The circuits are comprising capacitors, diodes and switches while being coupled with slits constituting the metasurface supercell. In time domain, the change in the incident frequency corresponds to changing the position of the DC source which initializes and flips the sign of the response. The average transmittance as a function of pulse period is represented while the response when the frequency is switched, can be dramatically controlled from the duration of the pulse. Measurements somehow validate the simulation result.*

The paper analyzes an interesting idea of the well-known frequency hopping translated into photonic communication with metasurfaces. However, it is not suitable for a publication at Nature Communications in its present form. In particular:

Response: We thank the reviewer for the feedback and constructive comments that have fully been considered on the revised version of the manuscript. We believe that thanks to these comments the paper has been further improved now.

Comment 1: *(A) The assessment of the quality of the communications should occupy a significant part of the paper. For example, a typical drawback of frequency hopping is the interference between the transmissive waves of neighboring frequencies. The authors should accompany their results with an evaluation of such a figure of merit.*

Response: Thank you for your valuable comment. We acknowledge the importance of assessing the communication quality in our study, and we have taken steps to address this concern. A new section has been incorporated into our work specifically dedicated to evaluating the communication performance of the metasurface. Within this section, as seen in the following figure, we conducted experiments to assess the interference between the transmissive waves of neighboring frequencies. To achieve this, deliberate interference signals were employed. The obtained results demonstrate the metasurface's robustness against interference from neighboring frequencies. This resilience can be attributed to the metasurface's characteristic of allowing high transmittance only for specific frequency bands utilized by the transmitted signal. Moreover, as suggested by the reviewer, our results quantitatively evaluate the transmittance reduction with various interference-to-signal ratios and with the original transmittance without any interference signals (please refer to Fig. 5f for this). By including this evaluation, we aim to provide a more comprehensive understanding of the metasurface's performance in the context of frequency-hopping spread spectrum communication, particularly in relation to narrowband interference mitigation. Below is the newly added Figure 5d-f with caption.

Fig. 1: (d, e) Spectrogram of (d) input and (e) output frequency-sequence signals corresponding to sequence #1 in Fig. 4d with double-band interference signals at $f_4 = 2.9$ GHz and $f_5 = 3.6$ GHz. (f) Transmittance reduction with different interference-to-signal ratios (I/Ss). The transmitted energies of frequency-sequence signals and interference signals are compared to the transmitted energy of sequence #1 shown in Fig. 4d. Here, the interference signal magnitude (f_4 and f_5) is swept following the interference-to-signal ratio (I/S) from -14 dB to 0 dB.

In addition to the aforementioned evaluation of interference resilience, we have conducted further experiments involving the transmission of realistic communication signals containing binary data. This analysis allows us to investigate the performance of the proposed metasurface in selectively transmitting a predetermined frequency-hopping sequence between a transmitter and receiver. Specifically, we examined the relationship between the bit error rate (BER) and the signal-to-noise ratio (S/N) when utilizing the metasurface for communication with the binary phase shift keying (BPSK) modulation scheme and a frequency-hopping carrier. When the metasurface is employed with the correct frequency-hopping sequence, i.e., consistent with the sequence embedded within it, the BER remains low, indicating successful transmission. However, if the frequency sequence of the signal differs from the sequence predefined by the metasurface, the BER increases, reflecting errors in data transmission. It is noteworthy that the metasurface's selectivity for frequency-hopping signals is an inherent feature derived from its physical properties. This selectivity persists even when both the transmitter and receiver adopt the same frequency sequence for modulation and demodulation processes. Hence, the metasurface can provide physical-level information encryption, operating based on the frequency sequence of the incoming signal. By conducting these experiments and highlighting the metasurface's performance in terms of both interference resilience and BER-S/N relationship, we aim to provide a comprehensive understanding of its capabilities in realistic frequency-hopping spread spectrum communication scenarios. Below is the newly added Figure 5g-i with caption.

Fig. 2: (g) Diagram block of communication scenario using the proposed metasurface and the BPSK modulation scheme. The time length for each bit T_b is set to $T_b = 10$ ns, while the time slot for each frequency carrier is 300 ns (corresponding to 30 bits). (h) Examples of the received BPSK signals comparing two frequency sequences (left) before and (right) after implementing additive white Gaussian noise. (i) BER vs. S/N analysis using two different frequency sequences.

Regarding the difference in BER between the two frequency sequences depicted in Figure 5i, we have conducted additional measurements to transmit a realistic binary image. The measurement results, specifically related to the transmission of a binary image using the proposed metasurface configuration, have been included as Figure S20. The output demonstrates an image with fewer errors when the correct frequency sequence (f_1, f_2, f_3) is utilized, in contrast to the output obtained from the incorrect frequency sequence (f_3, f_2, f_1).

Figure S20: Example of data transmission (logo mark of Nagoya Institute of Technology) using the diagram block shown in Fig. 5g. The SNR is varied from -20 to 0 dBm. Two frequency sequences are used for the carrier, similar to the scenario explained in Fig. 5c.

Revised Parts: To reflect the above discussion, we have added an entirely new section discussing the quality of wireless communication in Subsection 3.4. In the Supplementary Note 5, we have added a discussion on the related measurement results in which the transmission of a binary image using the proposed metasurface configuration has been conducted. Moreover, we have added a discussion about the potential applications of the metasurface including physical level security enabled by the metasurface, in the third paragraph of Section 4.

Comment 2: (B) *The authors should examine the possibility of expanding their paradigm to phase hopping and provide the reader with a list of pros and cons against their current frequency hopping.*

Response: First, please allow us to highlight the distinction between frequency hopping spread spectrum (FHSS) and Direct Sequence Spread Spectrum (DSSS). FHSS entails the deliberate alteration of the carrier frequency according to a known pattern shared by the transmitter and receiver. In contrast, DSSS modifies the phase of the signal using pseudo-random codes while maintaining a fixed carrier frequency within a specific band. This results in better inherent resistance to narrowband interference for DSSS signals compared to FHSS signals. Moreover, since only one carrier frequency is used in DSSS, its synchronization with the pseudo-random code is generally simpler to realize by employing a digital signal processing procedure. This is in contrast with FHSS, which requires synchronization between several analogue carrier frequency signals at the receiver.

From the perspective of hardware implementation, our metasurface with frequency-sequence selectivity better suits the implementation of FHSS technique. Our metasurface can serve as a passive device consisting of multiple unit-cells designed to passively couple to a predetermined frequency sequence. We agree with the reviewer that the same concept may be expanded to realize a metasurface that also work for a DSSS communication platform. In this new concept, the metasurface would be responsible to perform the correlation between the received signal and its locally generated replica of the pseudo-random codes. Ideally, the process should be conducted passively. However, this requires a digital signal processing which necessitates direct current sources. These challenges could be addressed in future studies exploring the use of metasurfaces with DSSS signals.

Revised parts: To discuss the possibility to expand the current metasurface platform for DSSS signals, we have added a discussion in the first paragraph of Section 4 as follows,

Moreover, although our concept appears analogous to the principle of the frequency-hopping spread spectrum (FHSS), it is also conceivable to design metasurfaces with a phase-hopping technique to conform to the direct sequence spread spectrum (DSSS). In such a scenario, the metasurfaces would incorporate phase modifications while maintaining a consistent carrier frequency within a specific frequency band. This approach may offer enhanced immunity to narrowband interference and facilitate simpler synchronization techniques, which are based on only single frequency operation. However, the metasurfaces need to be sensitive to phase variations to accurately replicate the pseudorandom phase coding in the transmitted signal, necessitating further investigation in future studies.

Comment 3: C) *The authors may discuss the connection of their concept with other metasurface-based communications setups involving secure transmission [1,2].*

[1] Metasurface-Coated Devices: A New Paradigm for Energy-Efficient and Secure 6G Communications, IEEE Vehicular Technology Magazine, 2021.

[2] Metasurface-Assisted Wireless Communication with Physical Level Information Encryption, Advanced Science, 2022.

Response: Thank you for providing the valuable reference and suggesting the connection with other metasurface-based communications setups involving secure transmission. We appreciate your interest in exploring the applications of our concept in the context of physical layer security. As mentioned earlier, our metasurface can provide physical-level information encryption based on the frequency sequence of the incoming signal. This characteristic can be utilized in communication systems that employ beamforming techniques, as discussed in the above-mentioned references [1, 2]. In beamforming-based physical layer security, the typical wiretap channel model involves one transmitter, one legitimate receiver and one eavesdropper. The secrecy capacity, which is a performance metric of such physical layer security, is determined by the difference between the capacity of the legitimate channel and that of the eavesdropper channel. While our current proposed scheme does not directly create a beam or null for physical layer security purposes, it can still contribute to the overall concept if reflection/transmission phase is changed over a two-dimensional plane. Note that similar works of realizing beamforming metasurfaces depending on the incoming pulse-width has been recently reported in our works for cloaked antennas and reconfigurable intelligent surfaces (RISs) (see Refs. [R1] and [R2]). The proposed metasurface, which provides frequency-hopping selectivity, can be used to enhance physical layer security by directing the beam towards the legitimate receiver and suppressing it towards the eavesdropper, depending on the utilized frequency-hopping sequences. However, it should be noted that further research would be needed to realize beamforming techniques specifically linked with the frequency-hopping selectivity of our metasurface. Potential avenues for future work could involve exploiting the periodicity of the proposed metasurface within the supercell to engineer diffraction into various angles, thereby enabling controlled beamforming. Another approach could involve utilizing a multi-layer structure, with one layer acting as a beamformer while our metasurface remains in the other layer. However, it is important to emphasize that these studies go beyond the scope of our currently presented work, but they offer promising directions for future investigations in the field of metasurface-based physical layer security.

[R1] Ushikoshi, D. et al. Pulse-driven self-reconfigurable meta-antennas. Nat Commun 14, 633 (2023).

[R2] Fathnan, A. A. et al. Unsynchronized Reconfigurable Intelligent Surfaces With Pulse-Width-Based Design. IEEE Trans. Veh. Technol. PP, 1–6 (2023) (early access).

Revised parts: In the revised manuscript, we have added a discussion about the potential applications of the metasurface including physical level security enabled by the metasurface, in the third paragraph of Section 4, including citations to Refs. [1] and [2], [R1] and [R2] above.

Comment 4: (D) The whole structure is one-dimensional. However, fields diffuse across all directions. How the analysis is modified if two-dimensional waves are utilized as happening in other memory structures [3,4]?

[3] Reconfiguring structured light beams using nonlinear metasurfaces, *Optics Express*, 2018.

[4] Angular memory of photonic metasurfaces, *IEEE Transactions on Antennas and Propagation*, 2021.

Response: We appreciate the reviewer's valuable suggestion regarding the analysis of the metasurface in the context of two-dimensional waves, as observed in [3] and [4]. While we agree that investigating the behavior of the metasurface in two-dimensional wave configurations, such as optical angular momentum or angle-dependent operations, is an interesting avenue for future research, we would like to clarify that our current study primarily focuses on the confirmation of frequency-hopping wave engineering using the proposed metasurface, which provides more selectivities than what the existing linear frequency concept does. However, to address this concern and demonstrate the versatility of our metasurface, we conducted additional simulations of the metasurface in a two-dimensional free-space configuration using horn antennas. These simulations aimed to show that our metasurface is not limited to rectangular waveguide applications (one-dimensional) and can also function effectively in a two-dimensional setting. As seen in the newly added Figure S9 below, the results obtained from these simulations align with those obtained from the rectangular waveguide setup, confirming the applicability of the metasurface in a two-dimensional configuration.

Figure S1: Simulation of the metasurface of Fig. 3a using the free-space configuration. (a) A pair of two horn antennas are used as a transmitter and a receiver in the electromagnetic simulation. The metasurface is deployed in a metallic conductor plate to narrow the metasurface area (lower the computational resources needed) while preventing the direction propagation of the signal between the transmitter and the receiver. The design parameters are shown in Table S6. (b) Circuit schematic integrating the electromagnetic simulation result. Two terminals are connected to the transmitter and receiver ports, and the remaining terminals are connected to diode bridges including inductors and resistors with parallel capacitors. (c) Simulation results of the cosimulation method. The top panel shows the transmitted power at the receiver when the transmitter generates a signal of 20 dBm. The bottom panel shows a transient transmittance that is obtained after normalization to the transmitted power when

the metasurface is absent. Comparison to the waveguide simulation (the blue line and the same as Fig. 3d) shows a good agreement between both configurations, indicating the possibility of extending the proposed metasurface to free-space application scenarios.

Revised parts: We have added a relevant discussion about the possibility of using the proposed metasurface in a free-space scenario in the second paragraph of Section 4, in which both References [3] and [4] above were cited. Detail of the simulation for the free-space scenario have been added in Supplementary Note 4 of Supplementary Information showing the newly added Figure S9 and the corresponding Table S6.

Responses to Reviewer 2

Comment 0: *In the manuscript “Frequency-Hopping Wave Engineering with Metasurfaces” the authors utilized diode bridge rectifiers in the meta-atoms to enable designed transient responses, which further improve the transmittance of EM waves.*

Response: Thank you very much indeed for your valuable feedback and insightful comments that have markedly enhanced the quality of our manuscript. We appreciate your recognition of the utilization of diode bridge rectifiers in our metasurface to enable transient response which indeed improved the transmittance of the metasurface and added a new degree of freedom based on frequency sequence. Please read our following responses that carefully addressed your important comments and suggestions.

Comment 1: *However, this phenomena only exist for a short time, let alone the transmittance is even dramatically varying. All these drawbacks make it hard to be used in modern electronic systems. Therefore, unless the authors clearly the application scenarios of such kind of device more clearly, I could not recommend it be published in Nature Communications. In addition, there are some other suggestions.*

Response: We appreciate your concerns regarding the transient nature of the metasurface and its applicability in modern electronic systems. We would like to clarify that the transient response utilized in our metasurface does not limit its operation to a short duration. The transient response is specifically utilized to enable direct-current coupling between resonances in different frequency bands, allowing for frequency-hopping selectivity over longer signals. This is evident from the presented Figure 4d in which the frequency selectivity appears even for longer waveform around 10 μs . To better clarify these results, we have also added spectrograms of the input and output signals and presented them as newly added panels 4e and 4f.

Fig. 3: Experimental demonstration of engineering wave propagation in accordance with frequency sequence. (a) Equivalent circuit system with variable resistors. The variable resistance values change in the time domain depending on the incoming frequency sequence. In the time domain, the incident frequency is repeatedly changed in a particular sequence, which determines how the position of the DC source is moved. (b) Specific circuit system. JFETs are included and biased by other circuits. (c) Measurement sample designed to operate at three frequencies. (d) Measured transient transmittances during the initial period and in the steady state (left) and their averages (right). f_1 , f_2 and f_3 are 3.3, 3.9 and 2.5 GHz, respectively. During the initial period, the number of distinct frequency sequences is determined by $N!$, where N represents the number of frequency channels available. In contrast, the number of distinct frequency sequences is reduced to a circular permutation of N , namely, $(N-1)!$. (e) Normalized spectrogram of the input and output signals (left and right, respectively) using the frequency sequence of f_1 , f_2 and f_3 . (f) Normalized spectrogram of the input and output signals (left and right, respectively) using the frequency sequence of f_3 , f_2 and f_1 . (e) and (f) correspond to sequences #1 and #6 in (d), respectively.

Furthermore, in the revised manuscript, we have included application scenarios of the metasurface for wireless communication to provide a clearer understanding of its potential. Specifically, we demonstrate that the metasurface can differentiate between two distinct wireless communication signals based on their frequency-hopping sequences, as shown in the BER vs. S/N analysis results of new Fig. 5. The analysis was performed using approximately 20 kbit data, where each bit had a duration of $T_b = 10$ ns, resulting in a total signal length of 0.2 milliseconds. This significantly exceeds the time constant of the transient circuit used, which is on the order of hundreds of nanoseconds. The following figure represents the newly added Figure 5.

Fig. 4: Experimental demonstration of the metasurface towards realistic communication scenarios. (a, b) Spectrogram of (a) input and (b) output chirp signals sweeping the oscillation frequency during each 300-ns time slot. The centre frequency in each time slot hops between f_1 , f_2 and f_3 , which correspond to sequence #1 in Fig. 4d. In this example, the chirp signal is swept by 0.4 GHz in each time slot. (c) Transmittance reduction with different sweep frequency ranges. The transmitted energies of chirp signals using different frequency sequences are compared to the transmitted energy of sequence #1 shown in Fig. 4d. (d, e) Spectrogram of (d) input and (e) output frequency-sequence signals corresponding to sequence #1 in Fig. 4d with double-band interference signals at $f_4 = 2.9$ GHz and $f_5 = 3.6$ GHz. (f) Transmittance reduction with different interference-to-signal ratios (I/Ss). The transmitted energies of frequency-sequence signals and interference signals are compared to the transmitted energy of sequence #1 shown in Fig. 4d. Here, the interference signal magnitude (f_4 and f_5) is swept following the interference-to-signal ratio (I/S) from -14 dB to 0 dB. (g) Diagram block of communication scenario using the proposed metasurface and the BPSK modulation scheme. The time length for each bit T_b is set to $T_b = 10$ ns, while the time slot for each frequency carrier is 300 ns (corresponding to 30 bits). (h) Examples of the received BPSK signals comparing two frequency sequences (left) before and (right) after implementing additive white Gaussian noise. (i) BER vs. S/N analysis using two different frequency sequences.

Also, the measurement results related to the above figure is shown in Figure S20, where transmission of a binary image using the proposed metasurface configuration has been conducted. The following figure is the measurement results shown in Figure S20.

Figure S20: Example of data transmission (logo mark of Nagoya Institute of Technology) using the diagram block shown in Fig. 5g. The SNR is varied from -20 to 0 dBm. Two frequency sequences are used for the carrier, similar to the scenario explained in Fig. 5c.

While acknowledging the reviewer's concerns regarding the limited transmittance exhibited by the proposed metasurface, it is essential to underscore the availability of various approaches to improve its performance. For instance, in subwavelength metallic slit structures, there is the tradeoff relation between a Q factor and its effective electrical size, which related to the energy dissipation and transmission spectrum of our metasurface. The observed low transmittance in our metasurface at present can be attributed to the relatively high Q factors caused by the small electrical length of the slit structure, as well as resistive losses introduced by the dielectric substrate. We have conducted simulations using a simplified equivalent transmission line model to investigate this aspect, and the result is summarized in Figure S10 (also shown in the next page). By intentionally introducing inductive and resistive components of the dielectric substrate (L_{add} and R_{add}), we observe a reduction in transient transmittance proportional to the resistance value and the resonance's Q factor. In such a configuration, resonances with low Q factors and low resistance are more favourable in maintaining high transmittance. However, it is crucial to consider that modifying Q factors in our proposed metasurface requires optimizing various factors, which may pose challenges in practical implementation. For example, commercially available dielectric substrates have fixed thickness and inherent loss, while the size of the slit structure cannot be arbitrarily altered considering its suitability for soldering circuits if lumped circuit components are manually soldered. However, there are alternative approaches, such as the use of on-chip technologies to minimize losses and add more rooms for geometrical variation of the slit structure.

Figure S2: Simplified equivalent transmission line model to represent the metasurface of Fig. 3a for analysing the contribution of the Q factor. (a) Equivalent circuit model connected to the transmission line for the single slit scenario (one unit cell). C_0 and L_0 represent the capacitive and inductive components of the metallic slit structure. L_{add} and R_{add} represent the inductive and resistive components contributed by the dielectric substrate. The circuit parameters used for the diode and the series inductor and resistor (inside the diode bridge) are the same as those used in Fig. 3. (b, c) Simulated transmittance from linear network analysis for the circuit shown in (a) with varying Q factors while considering (b) zero parasitic resistance $R_{\text{add}} = 0 \Omega$ and (c) nonzero parasitic resistance $R_{\text{add}} = 3 \Omega$ (see Table S7 for variation of C_1). (d) Equivalent circuit model connected to the transmission line for the slit scenario (one super cell). (e, f) Simulated transmittance from linear network analysis for the circuit shown in (d) with varying Q factors

while considering (e) zero parasitic resistance $R_{\text{add}} = 0 \Omega$, and (f) nonzero parasitic resistance $R_{\text{add}} = 3 \Omega$. (g, h) Nonlinear analysis of transient transmittance for four parallel resonance scenarios with switched frequency. Here, the input power is set to 10 dBm. f_1 , f_2 , f_3 and f_4 are 2.0, 2.5, 3.3 and 4.1 GHz, respectively. Simulated transient transmittance with varying Q factor while considering (g) zero parasitic resistance $R_{\text{add}} = 0 \Omega$ and (h) nonzero parasitic resistance $R_{\text{add}} = 3 \Omega$.

Revised parts: We have now revised the manuscript to more clearly demonstrate how our work was utilized beyond exploiting transient response in a short time duration. This includes revision into Figure 4 where Figures 4e and f were added showing the spectrogram of both input and output signals in two different frequency sequences. We have also included application scenarios of the metasurface for wireless communication to provide a clearer understanding of its potential, including the usage of a long waveform in the calculation of BER vs S/N analysis. A new section discussing the quality of wireless communication is now added in the manuscript as Subsection 3.4. In the Supplementary Note 5, we have added the measurement results related to transmission of a binary image using the proposed metasurface. Moreover, to discuss the possibility of increasing the transmittance, we have shown numerical analysis involving the equivalent transmission line model by in which the results is presented in the Supplementary Note 4 including an additional new Figure S10.

Comment 2: 1. *The definition of transmittance should be provided clearly. For example, in Figs. 3d, h and 4d, does it refer to the transmittance of the transient frequency?*

Response: Thank you for your valuable feedback. Now we have made a clear distinction between "transmittance" and "transient transmittance" within the manuscript to ensure that their specific contexts are properly conveyed. In the revised version of the manuscript, we define "transmittance" as a measure of the proportion to the total incident energy or signal that is transmitted through the structure over the whole signal duration. This term is used to quantify the transmission efficiency of the structure. On the other hand, "transient transmittance" is introduced to describe the time-varying behavior of the transmittance. It captures the dynamic changes in transmittance over time, particularly in response to transient or dynamic input signals (e.g., frequency-hopping signals in our case). By analyzing the transient transmittance, we gain insights into the temporal characteristics of the signal transmission process, allowing us to understand how the transmitting profiles change over time. These refined definitions of transmittance and transient transmittance enable a more precise and comprehensive description of the signal transmission phenomena discussed in our manuscript.

Revised Parts: To address the above comment, we have added a relevant discussion on the definition of transmittance and transient transmittance in the Methods section as shown below. In addition, we have edited figures both in the main manuscript and SI, to clearly show differentiation between "transmittance" or "transient transmittance".

Comment 3: 2. *In Fig. 2c, the descriptions of the blue and red lines are missing.*

Response: We thank the reviewer for the attention into detail. Now we have added texts next to each curve of Fig. 2c.

Revised parts: To address the above comment, we have now revised Fig. 2c as shown below.

Responses to Reviewer 3

Comment 0: *The paper proposes a kind of metasurface that can manipulate scattering waves according to the incidence frequency and pulse width in the designed bands. The metasurface loads different coupled transient circuits for different frequency and waveform selection, showing the scattering based on frequency combination and sequence. Numerical and experimental results demonstrate its efficiency of frequency-hopping wave engineering. My comments and suggestions are as follows:*

Response: We appreciate the positive feedback from the reviewer. In response to your valuable feedback, we have incorporated the necessary revisions into the manuscript to address any areas requiring clarification or improvement. We are grateful for your insights, as they have contributed to markedly refining our study and ensuring its scientific rigor.

Comment 1: *1. Some relative electromagnetic engineering works have been done, the new contribution over the work from the same group [1] is incremental. The design of rectifier-based circuit is a common technique in the antenna and propagation society. Much more comprehensive comparisons with the state of arts should be supplemented to highlight the novelty. Moreover, the measured transmission is very low, it is difficult to be employed in realistic applications. [1] D. Ushikoshi, R. Higashiura, K. Tachi, A. A. Fathnan, S. Mahmood, H. Takeshita, and H. Wakatsuchi, "Pulse-driven self-reconfigurable meta-antennas," Nature Communications, 14(1), 633, 2023.*

Response: Thank you for seeking further clarification. In response to the concern raised regarding the incremental contribution of our work compared to prior research, we have compiled a summary of the main contributions from each relevant work, including the reference mentioned by the reviewer as shown by the following table.

Table R1. Comparison between several relevant works on non-linear metasurfaces reported in the literature

Reference	Topic and novelty	Method
[R1] M. Barbuto, et al. "Waveguide components and aperture antennas with frequency-and time-domain selectivity properties." IEEE Transactions on Antennas and Propagation 68.10 (2020): 7196-7201.	Design of a waveguide filtering antenna with both frequency- and time-domain selectivity based on circular shape irises loaded by rectifiers and transient circuits.	Simulation
[R2] V. Constantinos, A. Sarsen, and A. Alu. "Angular memory of photonic metasurfaces." IEEE Transactions on Antennas and Propagation 69.11 (2021): 7720-7728.	Metasurface that produces hysteresis responses depending on the incident beam angle even at the same intensity profile.	Analytical
[R3] S. Vellucci, et al. IEEE Transactions on Antennas and Propagation 68.3 (2019): 1717-1725.	Metasurface as a mantle cloak that can conceal or reveal an antenna depending on the incident pulse waveform, however without the ability to differentiate simultaneous pulse incidences.	Simulation
[R4] D. Ushikoshi, et al. "Pulse-driven self-reconfigurable meta-antennas." Nature Communications 14.1 (2023): 633.	Metasurface that can direct the beam of a monopole antenna into certain angles depending on the incident pulse-width even with simultaneous incidences.	Simulation and experiment.
This work	Metasurface that has different responses depending on the sequence of a transmitted frequency-hopped signal.	Simulation and experiment

From the above table, it is evident that the current work differs from previous studies. Particularly, our work introduces a novel aspect of a nonlinear metasurface that exhibits distinct responses based on the frequency sequence of a transmitted signal. Such a new selectivity is realized by introducing coupling between unit

cells operating at different frequencies through the use of nonlinear circuit. This specific contribution has not been reported in any other works, indicating the non-incremental nature of our research. Furthermore, thanks to the latest revisions, we have clarified experimentally that the proposed metasurface can be usefully exploited for communications where selectively transmitted data can be obtained depending on the carrier's frequency-hopping sequence. Further research and development can be envisioned, encompassing contributions to the field of physical-level security, frequency difference sensing and reconfigurable intelligent surfaces.

Revised Parts: To address the above concerns, in the first paragraph of Discussion and Outlook section, we have explicitly mentioned that our metasurface presents distinct selection capability based on frequency sequence which is absent from previous works with additional citations including References R1 and R3.

Comment 2: *2. Fig. 3 (d), (h), and Fig. 4(d) show that the transmittance of designed supercells all less than 0.4, performing low efficiency. And some cases (for example incident switched-frequency is f_1 and f_2 in fig. 3 (d)) are even worse. The authors should have an explanation.*

Response: Thank you for your valuable comment. We appreciate your feedback regarding the low transmittance observed in Fig. 3 (d), (h), and Fig. 4 (d). In response to this concern, we conducted additional simulations to investigate the underlying reasons for the observed low efficiency. Using a simplified equivalent transmission line model, we intentionally introduced circuit components to account for the dielectric substrate that contains an inductor (L_{add}) and a dielectric loss (R_{add}). The result demonstrated that the Q factor can be adjusted, which leads to an increase in transmittance by reducing the Q factor and the loss (R_{add}). These simulation results, showcasing the tunability of the Q factor and its impact on transient transmittance, are shown in the next page and the Supplementary Information for further clarification. However, it is important to note that achieving an improved Q-factor in the experimental setup poses challenges due to certain design parameters related to, for instance, the substrate thickness and the structure's size. Moreover, in our experimental setup, we acknowledge that parasitic elements resulting from the diode's soldering process may have contributed to the additional decrease in transmittance. Nevertheless, we would like to emphasize that the primary functionality of our metasurface as a frequency-hopped selectivity remains intact. The newly added Fig. 5 demonstrates its effectiveness as a spatial filter in a realistic communication scenario, providing more specific filtering based on the frequency sequence. For example, in Fig. 5i, a contrasting BER was obtained from two received BPSK signals having two distinct frequency sequences, although the transmittances were around 20 % in sequence #1 and 10% in sequence #6. We believe that these findings reinforce the utility and practicality of our proposed metasurface for its intended purpose.

Figure S3: Simplified equivalent transmission line model to represent the metasurface of Fig. 3a for analysing the contribution of the Q factor. (a) Equivalent circuit model connected to the transmission line for the single slit scenario (one unit cell). C_0 and L_0 represent the capacitive and inductive components of the metallic slit structure. L_{add} and R_{add} represent the inductive and resistive components contributed by the dielectric substrate. The circuit parameters used for the diode and the series inductor and resistor (inside the diode bridge) are the same as those used in Fig. 3. (b, c) Simulated transmittance from linear network analysis for the circuit shown in (a) with varying Q factors while considering (b) zero parasitic resistance $R_{add} = 0 \Omega$ and (c) nonzero parasitic resistance $R_{add} = 3 \Omega$ (see Table S7 for variation of C_1). (d) Equivalent circuit model connected to the transmission line for the slit scenario (one super cell). (e, f) Simulated transmittance from linear network analysis for the circuit shown in (d) with varying Q factors while considering (e) zero parasitic resistance $R_{add} = 0 \Omega$, and (f) nonzero parasitic resistance $R_{add} = 3 \Omega$. (g, h) Nonlinear analysis of transient transmittance for four parallel resonance scenarios with switched frequency. Here, the input power is set to 10 dBm. f_1 , f_2 , f_3 and f_4 are 2.0, 2.5, 3.3 and 4.1 GHz, respectively. Simulated transient transmittance with varying Q factor while considering (g) zero parasitic resistance $R_{add} = 0 \Omega$ and (h) nonzero parasitic resistance $R_{add} = 3 \Omega$.

Revised parts: To address the above comment, we have included the simulation results clarifying the Q factor effect for transmittance in Supplementary Note 4 including by adding a new Figure S10.

Comment 3: 3. In the section 2, it is not very clear to me how the waveform selection is working, although there are some references are mentioned 29-33. The authors should explain its mechanism with more details.

Response: Thank you for your comment and feedback regarding the clarity of the waveform selection mechanism described in Section 2 of the manuscript. We appreciate your observation and have taken significant steps to address this concern. In response to your comment, we have made extensive revisions to Section 2 to provide a more detailed explanation of the waveform selection mechanism, in particular, by associating the waveform selection mechanism with the well-known classic transient phenomena. We have elaborated on the underlying principles and operational aspects, taking account of the references cited (29-33) to support and enhance the clarity of the explanation.

Revised parts: Please refer to the last two sentences of the first paragraph of Section 2 within the manuscript.

Comment 4: 4. In Fig. 4d, the measured average transmittance during the initial time period and in the steady state are not consistent in the cases of frequency sequence #2 and #3, while in other cases of frequency sequences are the same. The authors should have an explanation.

Response: Thank you for your comment and for pointing out the difference in Fig. 4d regarding the measured average transmittance during the initial time period. In Fig. 4d, the observed difference in transmittance during the initial time period for sequences #2 and #3, compared to other sequences, is due to the transient condition of the system. During this initial time period, sequence #2 exhibits a single resonance with high transmittance, while sequence #3 shows two resonances with high transmittance. This disparity arises because the system has not yet reached the steady state and is still undergoing transient behaviour. However, in the steady-state condition, the number of distinct sequences reduces to a circular permutation of N , specifically $(N - 1)! = 2$ in this case (with N being the number of the frequencies used). As a result, sequences #1, #2 and #3 all exhibit high transmittance with the same level. It is important to note that these three sequences are part of the same cycle of the frequency-hopping signal if the sequences are repeated. Similarly, sequences #4, #5 and #6 show the same transmittance but with a reduced level in the steady state because the waveforms appear the same if the three sequences repeatedly continue.

Revised parts: To address the above comment, we have revised Fig. 4 to more clearly demonstrate how the input and output signals vary by time and frequency. Figure 4 now have new panels showing the spectrogram of both input and output signals in two different frequency sequences (please refer to panels e and f). We have also revised the corresponding description in the first paragraph of Subsection 3.3.

Fig. 5: Experimental demonstration of engineering wave propagation in accordance with frequency sequence. (a) Equivalent circuit system with variable resistors. The variable resistance values change in the time domain depending on the incoming frequency sequence. In the time domain, the incident frequency is repeatedly changed in a particular sequence, which determines how the position of the DC source is moved. (b) Specific circuit system. JFETs are included and biased by other circuits. (c) Measurement sample designed to operate at three frequencies. (d) Measured transient transmittances during the initial period and in the steady state (left) and their averages (right). f_1 , f_2 and f_3 are 3.3, 3.9

and 2.5 GHz, respectively. During the initial period, the number of distinct frequency sequences is determined by $N!$, where N represents the number of frequency channels available. In contrast, the number of distinct frequency sequences is reduced to a circular permutation of N , namely, $(N-1)!$. (e) Normalized spectrogram of the input and output signals (left and right, respectively) using the frequency sequence of f_1 , f_2 and f_3 . (f) Normalized spectrogram of the input and output signals (left and right, respectively) using the frequency sequence of f_3 , f_2 and f_1 . (e) and (f) correspond to sequences #1 and #6 in (d), respectively.

Comment 5: *5. The paper suffers from bad English expression in general, and it makes me confused when I read the paper.*

Response: Thank you for your feedback regarding the English expression in the paper. We apologize for any confusion caused during your reading. In fact, the previous version of the paper underwent editing by professional English editors several times to improve the overall language quality. In addition, we have made changes on the manuscript to reflect review comments. Including these changes, the manuscript has been edited by a professional English editor again to enhance the clarity and readability. We have taken great care to ensure that the revised version is more understandable and effectively communicates the intended message. However, please let us know should you feel further editing is needed to more improve the English expression.

Revised Parts: Revisions have been made on the entire manuscript to improve the English quality and readability of the paper.

Comment 6: *6. Line 129, "Fig. 2b" should be "Fig. 3b", please the authors confirm it.*

Response: Thank you for bringing this to our attention. We appreciate your careful reading and have thoroughly reviewed the mentioned figure reference. After careful consideration and examination, we would like to confirm that in line 129, the reference to "Fig. 2b" is indeed correct as originally stated. We have revised the manuscript to clarify this reference and ensure accurate and consistent figure labeling throughout the paper.

Revised Parts: A revision has been made in the way Fig. 2b and Fig. 3b are mentioned in Subsection 3.2, as follows,

As seen in Fig. 2b, where a metasurface unit cell was related to a DC circuit, the supercell of Fig. 3a is associated with four independent DC circuits, each of which has a DC source activated by a different incident frequency..

Comment 7: *7. Line 372, "(g)" should be "(d)", please the authors confirm it.*

Response: We appreciate the reviewer's keen eye for detail. After careful verification, we confirm that in line 372 (caption of Figure 4), the reference "(g)" should indeed be corrected to "(d)" as suggested by the reviewer. We apologize for the error in the original manuscript and appreciate the reviewer's diligence in catching this mistake.

Revised parts: In the caption of figure 4, (g) has been corrected to (d)

Reviewers' Comments:

Reviewer #1:

Remarks to the Author:

The authors have properly addressed the raised points during the first review round. This reviewer recommends publication at Nature Communications.

Reviewer #2:

Remarks to the Author:

I have read through the revised paper, and found the current version still lacks enough novelty for the acceptance of Nature Communications. The critical problems of the revised papers can be found below:

1. The authors try to modify the metasurface transmission properties by tuning the incident frequency sequences. But what is the unique role of the metasurface? In my opinion, the common case is to fix the source and change the metasurface response. The motive of this paper is still confusing.
2. In Fig. 3d, it can be seen that the transient transmittance is enhanced by increasing the number of incident frequencies. The authors should give some insightful explanations instead of merely displaying the simulation results.
3. In Fig. 3d, the authors should further present the transmittance of each frequency component. It is not clear how the signals are influenced when passing the metasurface at different frequencies.
4. In the experiment of "the transmission of realistic communication signals", the results indicate that the BER is low with the correct frequency sequence and high with the improper ones. But what is the advantage of the current complicated communication system? I think the function of the metasurface is not obvious for the enhancement of the communication quality.

Reviewer #3:

Remarks to the Author:

In the response letter, the authors try hard to address my concerns.

As to novelty, the papers below [1]-[4] should also be considered, which involve similar function of frequency and pulse width selection.

[1] Bhattacharya, A., Dasgupta, B., & Jyoti, R. (2021). A simple frequency selective surface structure for performance improvement of ultra-wideband antenna in frequency and time domains. *International Journal of RF and Microwave Computer-Aided Engineering*, 31(11), e22857.

[2] Wang, Y., Min, X., Zhao, M., Yuan, H., Li, R., Hu, X., & Cao, Q. (2022). Design of a Frequency Selective Resonator With a Waveform Selective Passband. *IEEE Antennas and Wireless Propagation Letters*, 21(10), 2125-2129.

[3] Antony, A., & Dasgupta, B. (2023). Design and Analysis of a Frequency Selective Surface Loaded Bioinspired Antenna in Frequency and Time Domains. *Progress In Electromagnetics Research M*, 116.

[4] Wakatsuchi, H. (2015). Waveform-selective metasurfaces with free-space wave pulses at the same frequency. *Journal of Applied Physics*, 117(16).

In the comment 2, using the equivalent circuit can analyse transient transmittance of the unit-cell in principle, a similar solution is found to match the result of simulation or experiment. After analysis, the reduction of transient transmittance is mainly due to the parasitic inductor and resistance (lossy). And the resonance's Q factor is another important factor. But there is a certain distance between the

equivalent circuit model and metasurface structure, it is a realistic engineering problem. Although the authors added a new successful experimental of BPSK communication with frequency-hopping carrier under low transient transmittance condition, this drawback of the proposed metasurface is really existed, which will lead to energy loss in applications. Thus, I suggest that the author should improve the metasurface performance from structure itself, which is a very important point in your work. Besides, for the proposed complex but low efficiency communication scheme, are there any advantages comparing with other metasurface-based communication methods [5]-[8]?

[5] Hodge, J. A., Mishra, K. V., & Zaghloul, A. I. (2020). Intelligent time-varying metasurface transceiver for index modulation in 6G wireless networks. *IEEE Antennas and Wireless Propagation Letters*, 19(11), 1891-1895.

[6] Zhang, L., Chen, M. Z., Tang, W., Dai, J. Y., Miao, L., Zhou, X. Y., & Cui, T. J. (2021). A wireless communication scheme based on space-and frequency-division multiplexing using digital metasurfaces. *Nature electronics*, 4(3), 218-227.

[7] Chen, M. Z., Tang, W., Dai, J. Y., Ke, J. C., Zhang, L., Zhang, C., & Cui, T. J. (2022). Accurate and broadband manipulations of harmonic amplitudes and phases to reach 256 QAM millimeter-wave wireless communications by time-domain digital coding metasurface. *National science review*, 9(1), nwab134.

[8] Zhao, J., Yang, X., Dai, J. Y., Cheng, Q., Li, X., Qi, N. H., & Cui, T. J. (2019). Programmable time-domain digital-coding metasurface for non-linear harmonic manipulation and new wireless communication systems. *National science review*, 6(2), 231-238.

Responses to Reviewer 1

Comment 0: *The authors have properly addressed the raised points during the first review round. This reviewer recommends publication at Nature Communications.*

Response: We are genuinely grateful for your thorough review and positive recommendation for the publication of our manuscript. Your insights and constructive feedback have greatly contributed to enhancing the quality and clarity of our research.

Responses to Reviewer 2

Comment 0: *I have read through the revised paper, and found the current version still lacks enough novelty for the acceptance of Nature Communications. The critical problems of the revised papers can be found below.*

Response: Thank you for your thorough review of the revised paper. We appreciate the reviewer's feedback, and we have diligently taken steps to address these valuable comments. We believe that thanks to these comments the paper has been further improved now.

Comment 1: *The authors try to modify the metasurface transmission properties by tuning the incident frequency sequences. But what is the unique role of the metasurface?*

Response: Historically, artificially engineered subwavelength structures were intensively explored for more than half a century as frequency selective surfaces,¹³ metamaterials^{14,15} and metasurfaces.¹⁶ which successfully produced a wide breadth of applied devices and systems in the domains of antenna design, wireless communications,^{22,25} sensing,⁴⁹ imaging,⁵⁰ wireless power transfer¹¹ and bio/medical applications.⁵¹ However, the electromagnetic response of these structures is governed by frequency as they have frequency selectivity, hence limiting the degrees of freedom for electromagnetic wave manipulation. In recent years, efforts have been made to expand metasurfaces' capacity through additional parameters, thereby enabling multi-channel modulation even when the same frequency was used. For example, Reference [R4] introduced a metasurface capable of distinct wavefront manipulation in four circularly polarized output channels through the control of chirality-assisted phase response. In another instance, polarization control has been utilized to realize additional channels [R1-R3] for distinct holographic images associated with a right or left circular polarization, enhancing information encryption capacity in a photonic system [R1]. In Reference [39], spatial variation of waveform-selective metasurfaces around an omnidirectional antenna has enabled selectivity under simultaneous incidence hence increasing a channel number based on pulse width of the incident radio signals. As opposed to these recent structures, our metasurface plays a unique role in wave filtering based on frequency sequences even with the same frequency resources. Particularly, the metasurface demonstrates diverse scattering patterns depending on the incident frequency sequences which results in a significant expansion of available frequency channels following a factorial number ($N!$ with N being the frequency number). This has not been reported in any previous studies on the topic of channel enhancement based on additional degrees of freedom.

We can more appreciate this point if we compare in detail what channel number can actually be achieved in other devices with different methods of channel enhancement [R1,R2,R3,R4]. For instance, despite many efforts in [R1] to realize a two-dimensional metasurface with distinct responses for both right and left circular polarizations, only three channels were achieved in addition to three frequencies used (total channel number= 6). In contrast, in our metasurface, by utilizing only three frequencies ($N=3$) with the same polarization, the number of channels involving frequency sequence is $N! = 3! = 6$. By adding one more frequency ($N=4$), the channel number can drastically increase to $N! = 4! = 24$. Note that none of the previous works have utilized frequency sequences for the method of transmittance tuning and channel enhancement, which strategically positions our work distinct from prior studies. Moreover, as seen in the history of artificially engineered structures, our metasurface has potential to develop new types of applied devices and systems. For example, the metasurface can be utilized to realize secure electromagnetic buildings or smart shielding [R5-R6], which leads to highly confined radio signals in designated areas of interest, enhancing wireless security. The frequency hopping selectivity can also be seamlessly integrated into other devices to realize physically unclonable functions (PUF) [R7-R8] and wireless internet of things (IoT) tags [R9-R10]. Particularly, in the context of wireless IoT tags, this additional degree of freedom, represented by the frequency sequence, enhances the available channel capacity. Consequently, more devices can be accommodated using IoT tags with distinct frequency sequences.

To clearly address this review comment and reflect the above responses, we have added one paragraph in the introduction section discussing previous studies on the topic of channel enhancement based on additional degrees of freedom. In addition to new citations of Refs. [R1-R4] which exploited polarization and chirality for enhancing selectivity or channel number, we have also included discussion on Ref. [39], which exploit pulse width of the incident wave. In addition to some additional changes, section 1 was largely revised as follows:

From an engineering perspective, the assignment of frequency resources is based on a *linear* frequency concept and maximized in accordance with the frequency bandwidth resolved.⁴⁻⁶ In other words, the number of available frequency channels is proportional to the number of frequency slots used (Fig. 1a). However, the frequency channel number potentially increases if electromagnetic materials and their applied devices can behave differently within the same frequency resources. In particular, since controlling material response is limited at a single frequency, achieving more degrees of freedom over more than one frequency will be important to break the conventional linear frequency concept and increase the frequency channel number and channel capacity⁷.

Historically, artificially engineered subwavelength structures were intensively explored for more than half a century as frequency selective surfaces (FSSs),⁸ metamaterials^{9,10} and metasurfaces,¹¹ which successfully produced a wide breadth of applied devices and systems in the domains of, for instance, antenna design,¹² wireless communications,^{13,14} sensing,¹⁵ imaging,¹⁶ wireless power transfer¹⁷ and bio/medical applications.¹⁸ However, as mentioned above, the electromagnetic response of these structures is also governed by frequency as they have frequency selectivity, hence limiting the degrees of freedom for electromagnetic wave manipulation. In recent years, efforts have been made to expand metasurfaces' capacity through additional parameters, thereby enabling multi-channel modulation even at the same single frequency. For example, a metasurface was reported to be capable of distinct wavefront manipulation in four circularly polarized output channels through the control of chirality-assisted phase response [R4]. In another instance, polarization control has been utilized to introduce additional channels [R1-R3] for distinct holographic images associated with a right or left circular polarization, enhancing information encryption capacity in a photonic system [R1]. Moreover, spatial variation of waveform-selective metasurfaces deployed around an omnidirectional antenna has enabled the selectivity of pulse width even under simultaneous incidence.²³ Despite these efforts to exploit all intrinsic local properties such as frequency, amplitude, phase, polarization and pulse width, the linear frequency concept remained the same and limited the available channel number as long as systems were constrained by LTI conditions.

Moreover, we have modified a sentence within the first paragraph of section 4, where [R1] was cited.

While several relevant efforts on linear and nonlinear metasurfaces have reported advanced wave selectivities, such as polarization dependency [R1-R3], angular memory⁵⁹ and pulse-width dependency,^{23,44,47,56} none of the existing studies have exploited selectivity based on frequency sequences.

[R1] Huang, L., Mühlenbernd, H., Li, X., Song, X., Bai, B., Wang, Y. and Zentgraf, T., 2015. Broadband hybrid holographic multiplexing with geometric metasurfaces. *Advanced Materials*, 27(41), pp.6444-6449.

[R2] Dong, Fengliang, and Weiguo Chu. "Multichannel-independent information encoding with optical metasurfaces." *Advanced Materials* 31.45 (2019): 1804921.

[R3] Li, Z., Yu, S. and Zheng, G., 2020. Advances in exploiting the degrees of freedom in nanostructured metasurface design: from 1 to 3 to more. *Nanophotonics*, 9(12), pp.3699-3731.

[R4] Yuan, Y., Zhang, K., Ratni, B., Song, Q., Ding, X., Wu, Q., Burokur, S.N. and Genevet, P., 2020. Independent phase modulation for quadruplex polarization channels enabled by chirality-assisted geometric-phase metasurfaces. *Nature communications*, 11(1), p.4186.

[R5] Raspopoulos, M. and Stavrou, S., 2011. Frequency selective buildings through frequency selective surfaces. *IEEE Transactions on Antennas and Propagation*, 59(8), pp.2998-3005.

[R6] Trappe, W. ed., 2010. *Securing wireless communications at the physical layer* (Vol. 7). New York: Springer.

[R7] Yang, M., Ye, Z., Pan, H., Farhat, M., Cetin, A.E. and Chen, P.Y., 2023. Electromagnetically unclonable functions generated by non-Hermitian absorber-emitter. *Science Advances*, 9(36), p.eadg7481.

[R8] Yang, M., Zhu, L., Zhong, Q., El-Ganainy, R. and Chen, P.Y., 2023. Spectral sensitivity near exceptional points as a resource for hardware encryption. *Nature Communications*, 14(1), p.1145.

[R9] Tashiro, M., Fathnan, A.A., Sugiura, Y., Uchiyama, A. and Wakatsuchi, H., 2022. Metasurface-inspired maintenance-free Internet of things tags characterised in both frequency and time domains. *Electronics Letters* 58, 25, 937-939.

[R10] Kortuem, G., Kawsar, F., Sundramoorthy, V. and Fitton, D., 2009. Smart objects as building blocks for the internet of things. *IEEE internet computing*, 14(1), pp.44-51.

Revised Parts: This important comment and discussion are reflected at the nineth and eleventh lines of abstract, the first to third paragraphs of section 1, the first paragraph of section 4 and the sixth line from the end of section 4.

Comment 2: *In my opinion, the common case is to fix the source and change the metasurface response. The motive of this paper is still confusing.*

While we acknowledge that it is possible to fix the source and tune the metasurfaces using external control entity (e.g., through biasing of diodes), this study presents a different perspective. In contrast to the active metasurface paradigm mentioned by the reviewer, our passive metasurface was deliberately designed to exploit the variation in the source signal through introduction of self-tunability. There are two reasons to support our paradigm. Firstly, in wireless communications, variation of source signal is commonly used. For example, modulation to the amplitude, phase and frequency of the source signals has been utilized to enhance the capacity, range and quality of wireless communications. Specifically, frequency hopping spread spectrum (FHSS), which rapidly changes the carrier frequency, is one example of advanced modulation techniques which we exploit in this work. In fact, this modulation scheme is widely used as Bluetooth products. Secondly, utilizing the self-tunability to modify the metasurface response results in a simpler realization of wireless communication environment. Specifically, our metasurfaces do not require any external biasing lines and precise time synchronization with transmitting sources thanks to the self-tunable mechanism. Therefore, our metasurface was designed based on the paradigm that source signals can be exploited to change the metasurface responses and vary the wireless environment.

To summarize the preceding discussion, we have extended the introduction section by creating a final paragraph (paragraph 4). In this new paragraph, we have integrated the last two sentences from the previous paragraph 3, resulting in the following content:

In this work, we introduce a new approach to increase electromagnetic wave selectivity or channel number using passive metasurfaces, which breaks the conventional linear frequency concept and overcomes the limitation imposed by classic LTI systems. Unlike previous metasurfaces that achieve variable scattering properties in accordance with frequency, polarization, amplitude, phase or pulse width of incident waves, we propose the use of multiple frequencies or, more specifically, *frequency sequences* as a new degree of freedom (Fig. 1b). Additionally, unlike active tunable metasurfaces that rely on external control systems, we develop passive yet variable metasurfaces that operate differently in response to the pulse width of incident waves but without any electrical biasing systems. Consequently, our metasurfaces have the potential to significantly expand the number of frequency channels while providing symbol-level synchronization, thanks to their passive and self-tunability operation. The proposed concept of obtaining distinct responses with variation in frequency sequences is akin to frequency hopping, the spread-spectrum modulation scheme used for Bluetooth,⁴⁰ and is utilized here as an analogue filter to spatially control electromagnetic waves.

Revised Parts: Please refer to the last paragraph of the Introduction section.

Comment 3: *In Fig. 3d, it can be seen that the transient transmittance is enhanced by increasing the number of incident frequencies. The authors should give some insightful explanations instead of merely displaying the simulation results.*

Response: We thank the reviewer for seeking clarification on this particular point. The transient transmittance is enhanced by increasing the number of incident frequencies, which relates to a recovery time to restore the inductor voltage to its initial condition. More complete explanation is as follows: when a wave impinges on the metasurface, the rectified energy is stored as magnetic energy within the inductor that generates an electromotive force (EMF) within the circuit. When the wave ceases (the pulse stops), the inductor discharges the stored energy in the form of induced voltage, namely, a back EMF that circulates current within the circuit despite the absence of the original source (incident wave). In the bottom right of Fig. 3b, this induced voltage is shown as a negative voltage slowly increasing to zero. *The time needed for the inductor voltage to return to zero is called the recovery time.* If another wave using the same frequency comes

during this recovery time, the metasurface's transmittance cannot be maximized since the inductor accumulates the magnetic energy, causing a continuous flow of current and hence effectively reducing the transmittance of the slit metasurface. However, if another wave occurs with a different frequency, the magnetic energy does not accumulate due to the independence of other rectifier circuits.

In our metasurface seen in Fig. 3a, one particular meta-atom rectifies an incident wave of a particular operating frequency. Therefore, when the frequency of a pulse is regularly changed, energy charge and discharge repeatedly occur in inductors with properly designed frequencies. By increasing the number of frequencies used, there will be additional time to repeat the same pulse again. Therefore, the inductor voltage further approaches zero, which indicates that the slit metasurface improves transient transmittance. For this reason, as seen in Fig. 3d, the transient transmittance is enhanced by increasing the number of incident frequencies.

To clarify the above points, we ran additional simulations using repeated pulses with various duty cycles, which shows how the recovery time affects the transmittance. The result is now added as Figures S6 h-i, and the corresponding discussion is now added to Supplementary Note 4 as follows.

Furthermore, to see how the recovery time and the transmittance are related to the pulse period, additional simulations are performed in Figure S6h. Here, the pulse width is fixed at 100 ns, while the duty cycle is set to either 0.25, 0.5 or 0.75 (i.e., the pulse period of 400 ns, 200 ns or 150 ns). As a result, the transient transmittance is maximized when the duty cycle is 0.25 (or the pulse period of 400 ns). This is because the inductor voltage requires a recovery time of about 300 ns to approach its original voltage value as seen in Figure S6i (see that the voltage becomes zero in approximately 300 ns after the pulse duration of 100 ns). Therefore, when the pulse period is smaller than 400 ns, the metasurface's transient transmittance cannot be maximized since the inductor voltage is not fully restored yet. For the multi-resonance case, as seen in Fig. 3d, the transient transmittance is enhanced by increasing the number of incident frequencies since it extends the pulse period and the recovery time to restore the inductor voltage.

Figure S6: (h) Time-domain profile of transient transmittance at 4.9 GHz with repeated pulses. The pulse width is fixed at 100 ns, while the duty cycle is set to either 0.25, 0.5 or 0.75. (i) Inductor voltage during and after a 100-ns pulse. In (h) and (i), RL and L are set to 10 Ω and 10 μH , respectively.

Revised Parts: Please refer to the newly added Figures S6 h-i and the corresponding discussion in the second paragraph of Supplementary Note 4.

Comment 4: *In Fig. 3d, the authors should further present the transmittance of each frequency components. It is not clear how the signals are influenced when passing the metasurface at different frequencies.*

Response: We thank the reviewer for the feedback and attention to detail. We have now included spectrogram plots to show the influence on each frequency component of the frequency switching case shown in Fig. 3d. Furthermore, for a fair comparison, we have also included the spectrogram plot of Fig. 3c where the fixed frequency case was used. The updated Fig. 3 is shown below. In the new spectrogram of Fig. 3c (bottom panel), we can see that for a single-frequency case using $f_1 = 2$ GHz, the maximum transmittance is around 0.045. However, when we periodically switch the frequency, the maximum transient transmittance increases to 0.4 as seen from the spectrogram of Fig. 3d (bottom panel). Note that this spectrogram is from the four-frequency case as shown by the purple curve in the top panel of Fig. 3d.

Fig. 1: Numerical and experimental demonstration of engineering wave propagation in accordance with frequency combinations. (a) Supercell of quadband waveform-selective metasurfaces. C_1 , C_2 , C_3 and C_4 are 0.1, 0.3, 0.6 and 1.1 pF, respectively. (b) Equivalent circuit system (left) and expected time-domain profiles (right). In the time domain, the incident frequency is repeatedly changed (top right), which corresponds to changing the position of the DC source (middle right). An inductor voltage can be restored to zero voltage while other frequencies are used (bottom right). (c) Simulated transient transmittance for single-frequency cases (top) and spectrogram of the transient transmittance for 2.0 GHz (bottom). (d) Simulated transient transmittance for switched-frequency cases (top) and spectrogram using all of the four frequencies (bottom). f_1 , f_2 , f_3 and f_4 are 2.0, 2.5, 3.3 and 4.1 GHz, respectively. (e) Average transmittance as a function of the entire pulse period. (f) Measurement sample designed to operate at two frequencies within a standard rectangular waveguide. (g) Measured transient transmittance for single-frequency cases. (h) Measured transient transmittance for the switched-frequency case. f_1 and f_2 are adjusted to 2.58 and 3.46 GHz, respectively. In these simulations and measurements, the input power is set to 10 dBm.

Revised Parts: Please refer to the revised Fig. 3 in the main manuscript in which new panels of spectrogram are shown in Figs. 3c and 3d.

Comment 5: *In the experiment of “the transmission of realistic communication signals”, the results indicate that the BER is low with the correct frequency sequence and high with the improper ones. But what is the advantage of the current complicated communication system? I think the function of the metasurface is not obvious for the enhancement of the communication quality.*

Response: We express our gratitude for the reviewer's valuable feedback and the request for a clarification of the metasurface's function. It is important to note that the diagram blocks representing the transmitter and receiver in Fig. 5g are adopted from the commonly used spread spectrum technique known as frequency-hopping spread spectrum (FHSS). Thus, we emphasize that this system is not as complex as the reviewer indicated. To clarify this point, we added the following sentence in the first paragraph of Subsec. 3.4,

It is important to note that the diagram blocks representing the transmitter and receiver in Fig. 5a are adopted from the spread spectrum technique widely known as frequency-hopping spread spectrum (FHSS).⁴⁰

To clarify the intended function of the metasurface in wireless communication especially related to the BER analysis, we have modified discussions in Subsec. 3.4. Firstly, please note that in Fig. 5g, the metasurface is positioned within the communication channel or inside the rectangular waveguide to demonstrate its ability to introduce selectivity based on the frequency sequence. As a result, we observe an improved Bit-Error Rate (BER) performance specifically in the case of the preferred frequency sequence “ $f_1f_2f_3$ ”, in contrast to the opposite sequence “ $f_3f_2f_1$ ”. This clearly indicates that *the metasurface functions as a spatial filter*, similar to the well-known concept of frequency-selective surfaces (FSSs) that preferentially transmit signals depending on the incident frequency. However, as opposed to classic FSSs, the proposed new functionality can benefit wireless communications in various ways. For example, FSSs have been used to improve security in indoor environment as selective windows of communication for transmitting a designed frequency while blocking other frequencies, as discussed in Ref. [R5-R6]. The same concept is applicable to our metasurface where not only single-frequency signals but even frequency-hopping signals with non-correct sequences are blocked. Alternatively, beyond its role as a spatial filter, the metasurface can be integrated into smart antennas which can benefit applications including physically unclonable functions (PUF) [R7-R9] and internet of things (IoT) tags [R10-R11]. In particular, in the context of wireless IoT tags, the additional selectivity of the frequency sequence enhances the available channel capacity so that more IoT tags can be accommodated within the same wireless network using profiles dependent on the frequency sequence.

Therefore, to clarify these points, we have modified the content of Section 3.4. This includes, firstly, changing the order of the paragraph by starting with a discussion of the BER measurement results (previously positioned as the last paragraph of Section 3.4). Secondly, we have also added one paragraph for the discussion of functionalities of the metasurface in wireless communication, which is now presented as the second paragraph of Subsec. 3.4.

The beginning of the first paragraph of Subsec 3.4 is now changed as follows,

Metasurfaces function as spatial filters, similar to the well-known concept of FSSs that preferentially transmit a signal depending on the incident frequency. In contrast to classic FSSs, here, our metasurface shown in Fig. 4 selectively transmits signals in accordance with the frequency sequence.

A newly added second paragraph of Subsec. 3.4 is written as follows,

Importantly, Fig. 5c demonstrated that BER performance is improved in the case of the preferred frequency sequence (f_1 , f_2 and f_3 or sequence #1), which clearly indicates that the metasurface works as a spatial filter. This novel functionality opens the door to numerous potential applications in wireless communications. For example, the concept of our metasurfaces can be utilized to realize secure electromagnetic buildings or smart shielding [R5-R6]. In this application, by placing a metasurface as a selective window for communication, signals are allowed to enter a building only if the frequency-hopping sequence is correct, while other signals are rejected even with the same frequency resources. Thus, our concept can contribute to confining confidential radio signal communications in designated areas of interest, enhancing wireless security. Furthermore, the concept of frequency-hopping selectivity can be used not only as spatial filters but also as smart antennas. Such an antenna design holds promising potential across various domains, including physically unclonable functions (PUFs) [R7-R9] and internet of things (IoT) tags [R10-R11]. In the context of IoT tags, the additional degree of freedom in accordance with the frequency sequence provides more identities (IDs) and thus permits the simultaneous use of more IoT tags within the same wireless network.

[R5] Raspopoulos, M. and Stavrou, S., 2011. Frequency selective buildings through frequency selective surfaces. *IEEE Transactions on Antennas and Propagation*, 59(8), pp.2998-3005.

[R6] Trappe, W. ed., 2010. *Securing wireless communications at the physical layer* (Vol. 7). New York: Springer.

[R7] Gao, Y., Al-Sarawi, S. F. and Abbott, D., 2020. Physical unclonable functions. *Nature Electronics*, 3, 81–91.

[R8] Yang, M., Ye, Z., Pan, H., Farhat, M., Cetin, A.E. and Chen, P.Y., 2023. Electromagnetically unclonable functions generated by non-Hermitian absorber-emitter. *Science Advances*, 9(36), p.eadg7481.

[R9] Yang, M., Zhu, L., Zhong, Q., El-Ganainy, R. and Chen, P.Y., 2023. Spectral sensitivity near exceptional points as a resource for hardware encryption. *Nature Communications*, 14(1), p.1145.

[R10] Tashiro, M., Fathnan, A.A., Sugiura, Y., Uchiyama, A. and Wakatsuchi, H., 2022. Metasurface-inspired maintenance-free Internet of things tags characterised in both frequency and time domains. *Electronics Letters* 58, 25, 937-939.

[R11] Kortuem, G., Kawsar, F., Sundramoorthy, V. and Fitton, D., 2009. Smart objects as building blocks for the internet of things. *IEEE internet computing*, 14(1), pp.44-51.

Furthermore, the revised Fig. 5 is shown in the following,

Fig. 2: Experimental demonstration of the metasurface towards realistic communication scenarios. (a) Diagram block of communication scenario using the proposed metasurface and the BPSK modulation scheme. The time length for each bit T_b is set to $T_b = 10$ ns, while the time slot for each frequency carrier is 300 ns (corresponding to 30 bits). (b) Examples of the received BPSK signals comparing two frequency sequences (left) before and (right) after implementing additive white Gaussian noise. (c) BER vs. S/N analysis

using two different frequency sequences. (d, e) Spectrogram of (d) input and (e) output chirp signals sweeping the oscillation frequency during each 300-ns time slot. The centre frequency in each time slot hops between f_1 , f_2 and f_3 , which correspond to sequence #1 in Fig. 4d. In this example, the chirp signal is swept by 0.4 GHz in each time slot. (f) Transmittance reduction with different sweep frequency ranges. The transmitted energies of chirp signals using different frequency sequences are compared to the transmitted energy of sequence #1 shown in Fig. 4d. (g, h) Spectrogram of (g) input and (h) output frequency-sequence signals corresponding to sequence #1 in Fig. 4d with double-band interference signals at $f_4 = 2.9$ GHz and $f_5 = 3.6$ GHz. (i) Transmittance reduction with different interference-to-signal ratios (I/S s). The transmitted energies of frequency-sequence signals and interference signals are compared to the transmitted energy of sequence #1 shown in Fig. 4d. Here, the interference signal magnitude (f_4 and f_5) is swept following the interference-to-signal ratio (I/S) from -14 dB to 0 dB. In these measurements, the input power is set to 20 dBm.

Additionally, the BER results can be more improved if the metasurface transmittance is increased. To clarify this point, we have specifically designed and fabricated new metasurface samples that markedly improved transmittance performance. Specifically, the transmittance was nearly four times higher than that of the previous slit structure design. According to the equivalent circuit analysis reported for the previous review, the transmittance was increased by reducing the resonance Q factor. Therefore, in this new experiment, following the equivalent circuit approach, we translated a slit structure into a slightly different geometry in which the previous slit resonance was effectively replaced with lumped LC resonant circuits between metallic bars. This modification allowed us to readily adjust the Q factor of the resonant structure by changing the LC components. This experimental validation is summarized and presented in Figure S14, as seen in the following.

Fig. S14. Experimental demonstration of metasurface configuration based on lumped LC circuit resonators for increased transmittance. (a) Schematic of the metasurface with three metallic rows inside the rectangular waveguide (6 unit cells). (b) Schematic of the metasurface with four metallic rows inside the rectangular waveguide (9 unit cells). (c) The corresponding unit cell designs. (d, e) Measurement samples and their dimensions for (d) the three-metallic-row case and (e) the four-metallic-row case. The design parameters and circuit values used are shown in Table S12 and Table S13. (f) Frequency-domain profiles of the transmittance for 50-ns pulses. The result is compared to the slit configuration of Figure S12c. (g) Measured transient transmittance for switched-frequency cases. Here, the three frequencies used, i.e., f_1 , f_2 and f_3 , are changed in each structure to maximize transient transmittance in three different frequency bands. Specifically, for the 3-metallic-row case, f_1 , f_2 and f_3 are 2.69, 3.12 and 3.82 GHz, respectively, while for the 4-metallic-row case, f_1 , f_2 and f_3 are 2.83, 3.31 and 3.91 GHz, respectively. (h) Average transmittance for entire pulse period. In these measurements, the input power is set to 10 dBm.

In the third paragraph of Section 4, we have incorporated a corresponding discussion to explain this matter, as outlined below.

Despite the limited transmittance observed in the current metasurface design illustrated in Fig. 3e and Fig. 4d, we emphasize the availability of diverse strategies amendable to our metasurfaces for enhancing their transmittance, including optimizing the resonance's Q factors. Supplementary Note 6 provides information on how the transmitting characteristics can be further improved by reducing the resonance's Q factors, which helps our metasurfaces more efficiently work in specific application scenarios. To validate this approach, we specifically design such metasurfaces and present measurement results of transmittance that is markedly improved, compared to the slit structure design used above (see Supplementary Note 6 for details). It is worth noting that our effort to enhance transmittance by lowering the Q factor via manipulating the meta-atom configuration represents just one instance of such attempts. Potentially, other approaches are applicable to further improve transmittance, for instance, by changing material properties or by utilizing multi-layer metallic configurations.

We believe that this clarification elucidates the metasurface's significance within the field of wireless communication and information encryption. Moreover, we anticipate that its potential implications are appreciated by researchers in related fields like photonics and material sciences. Given the multidisciplinary nature of our study, we remain confident that Nature Communications provides the most fitting platform for disseminating our work, ensuring its impact across diverse scientific fields.

Revised Parts: Please refer to the entire Subsec. 3.4 and the third paragraph of Section 4 as well as newly added Figure S14, Tables 12 and 13 and the second paragraph of Supplementary Note 6 for the corresponding changes.

Responses to Reviewer 3

Comment 0: *In the response letter, the authors try hard to address my concerns.*

Response: We thank the reviewer for the further feedback and constructive comments that were fully addressed below. All of these comments were important to clarify the novelty of our study and further enhance the scientific importance.

Comment 1: *As to novelty, the papers below [1]-[4] should also be considered, which involve similar function of frequency and pulse width selection.*

[1] *Bhattacharya, A., Dasgupta, B., & Jyoti, R. (2021). A simple frequency selective surface structure for performance improvement of ultra-wideband antenna in frequency and time domains. International Journal of RF and Microwave Computer-Aided Engineering, 31(11), e22857.*

[2] *Wang, Y., Min, X., Zhao, M., Yuan, H., Li, R., Hu, X., & Cao, Q. (2022). Design of a Frequency Selective Resorber With a Waveform Selective Passband. IEEE Antennas and Wireless Propagation Letters, 21(10), 2125-2129.*

[3] *Antony, A., & Dasgupta, B. (2023). Design and Analysis of a Frequency Selective Surface Loaded Bioinspired Antenna in Frequency and Time Domains. Progress In Electromagnetics Research M, 116.*

[4] *Wakatsuchi, H. (2015). Waveform-selective metasurfaces with free-space wave pulses at the same frequency. Journal of Applied Physics, 117(16).*

Response: We express our gratitude to the reviewer for bringing up these relevant papers. Below we show a summary table for the above references comparing their novel contributions and methodologies.

Title	Novel contribution	Method
[1] A simple frequency selective surface structure for performance improvement of ultra-wideband antenna in frequency and time domains	Designing ultra-wideband antenna integrated with linear frequency selective surface based on patch geometry	Simulation, experiment
[2] Design of a Frequency Selective Resorber with a Waveform Selective Passband	Multifunctional surface with broadband absorption and waveform selectivity for free space waves	Simulation, experiment
[3] Design and Analysis of a Frequency Selective Surface Loaded Bioinspired Antenna in Frequency and Time Domains	Designing ultra-wideband antenna integrated with linear frequency selective surface based on lotus shape structure	Simulation, experiment
[4] Waveform-selective metasurfaces with free-space wave pulses at the same frequency	Waveform selective absorber for free space waves	Simulation, experiment

We see that Refs. [1] and [3] have contributed to the development of antennas integrated with frequency selective surfaces (FSS), while Refs. [2] and [4] contributed to the implementation of pulse-width-selective or waveform-selective metasurfaces in free space configurations. Particularly, Ref [2] have demonstrated multifunctional metasurfaces with broadband absorption and waveform selectivity for free space waves. However, it becomes apparent that our metasurface offers a unique contribution (i.e., varied transmittance depending on the incident frequency sequences) that distinguishes it from the metasurfaces presented in Refs. [1]-[4]. It is important to underscore that while Refs. [1] and [3] report transmittance variations based on incident pulse types, the underlying mechanism of those metasurfaces relies solely on linear responses, resulting in no inherent time variation within the structure. To further clarify this distinction, we have included these references in Section 4 (first paragraph) of the manuscript, and modify the discussion as shown in the following,

While several relevant efforts on linear and nonlinear metasurfaces have reported advanced wave selectivities, such as polarization dependency,²⁰⁻²² angular memory⁵⁹ and pulse-width dependency,^{23,44,47,56} none of the existing studies have exploited selectivity based on frequency sequences. In particular,

multifunctional metasurfaces with broadband absorption and waveform selectivity for free space waves have been reported but without frequency-sequence dependency.⁴⁶ Wideband FSSs have also been designed to distinguish different pulsed waveforms.^{60,61} However, the intrinsic mechanism was linear and had no time variation. Therefore, the linear frequency concept issue remained unchanged so far but was successfully overcome by our metasurfaces.

Furthermore, since Refs. [2] and [4] are also types of waveform-selective metasurfaces, we cited them in the first paragraph of section 2:

In particular, we use the recently proposed waveform-selective metasurfaces.⁴¹⁻⁴⁵ [2 and 4]

Revised Parts: We have included Refs. [1-4] at the twelfth line of the first paragraph of Section 4, contrasting their contributions with the metasurface presented here. Refs. [2] and [4] are also cited at the sixth line of the first paragraph of Section 2, as examples of waveform-selective metasurfaces.

Comment 2: *In the comment 2, using the equivalent circuit can analyse transient transmittance of the unit-cell in principle, a similar solution is found to match the result of simulation or experiment. After analysis, the reduction of transient transmittance is mainly due to the parasitic inductor and resistance (lossy). And the resonance's Q factor is another important factor. But there is a certain distance between the equivalent circuit model and metasurface structure, it is a realistic engineering problem. Although the authors added a new successful experimental of BPSK communication with frequency-hopping carrier under low transient transmittance condition, this drawback of the proposed metasurface is really existed, which will lead to energy loss in applications. Thus, I suggest that the author should improve the metasurface performance from structure itself, which is a very important point in your work.*

Response: Thank you very much for your valuable feedback and the opportunity to further improve our metasurface performance. We have fully considered your suggestion and specifically designed and fabricated new metasurface samples. As a result, we have achieved a significant improvement with the transmittance increased nearly four times higher in the frequency combination case. According to the equivalent circuit analysis conducted previously, the increase in transmittance can be realized by reducing the resonance Q factor. Therefore, in this experiment, for the realization of the metasurface, we translated a slit structure into a slightly different geometry in which the previous slit resonance was effectively replaced with lumped LC resonant circuits between metallic bars. This modification allowed us to readily adjust the Q factor of the resonant structure by changing the LC components. This experimental validation is summarized and presented in Figure S14, as seen in the following.

Fig. S14. Experimental demonstration of metasurface configuration based on lumped LC circuit resonators for increased transmittance. (a) Schematic of the metasurface with three metallic rows inside the rectangular

waveguide (6 unit cells). (b) Schematic of the metasurface with four metallic rows inside the rectangular waveguide (9 unit cells). (c) The corresponding unit cell designs. (d, e) Measurement samples and their dimensions for (d) the three-metallic-row case and (e) the four-metallic-row case. The design parameters and circuit values used are shown in Table S12 and Table S13. (f) Frequency-domain profiles of the transmittance for 50-ns pulses. The result is compared to the slit configuration of Figure S12c. (g) Measured transient transmittance for switched-frequency cases. Here, the three frequencies used, i.e., f_1 , f_2 and f_3 , are changed in each structure to maximize transient transmittance in three different frequency bands. Specifically, for the 3-metallic-row case, f_1 , f_2 and f_3 are 2.69, 3.12 and 3.82 GHz, respectively, while for the 4-metallic-row case, f_1 , f_2 and f_3 are 2.83, 3.31 and 3.91 GHz, respectively. (h) Average transmittance for entire pulse period. In these measurements, the input power is set to 10 dBm.

The new figure presented above serves the purpose of validating our previous hypothesis, which suggests that the transmittance of the metasurface can be enhanced by reducing the Q factor. As prompted by the reviewer, we initially conducted circuit simulations to assess this hypothesis. In this revision, we have supplemented our circuit simulation findings with experimental results, which clearly validates our hypothesis and the feasibility. In the third paragraph of Section 4, we have incorporated a corresponding discussion to elucidate this matter, as outlined below.

Despite the limited transmittance observed in the current metasurface design illustrated in Fig. 3e and Fig. 4d, we emphasize the availability of diverse strategies amendable to our metasurfaces for enhancing their transmittance, including optimizing the resonance's Q factors. Supplementary Note 6 provides information on how the transmitting characteristics can be further improved by reducing the resonance's Q factors, which helps our metasurfaces more efficiently work in specific application scenarios. To validate this approach, we specifically design such metasurfaces and present measurement results of transmittance that is markedly improved, compared to the slit structure design used above (see Supplementary Note 6 for details). It is worth noting that our effort to enhance transmittance by lowering the Q factor via manipulating the meta-atom configuration represents just one instance of such attempts. Potentially, other approaches are applicable to further improve transmittance, for instance, by changing material properties or by utilizing multi-layer metallic configurations.

Also, in the corresponding discussion in Supplementary Note 6, we have added the following paragraph,

To validate this circuit analysis results, we have conducted an experimental investigation employing a metasurface composed of metallic bars interconnected through lumped LC resonant circuits. The schematic is shown from Figure S14a to Figure S14c. This adjustment from the original metallic slit geometry to an LC interconnection scheme allows for convenient control of the Q factor of the resonant structure to achieve higher transmittance. To test the Q factor variation, the meta-atom periodicity in the x axis is altered as seen from Figure S14a to Figure S14b. Within the rectangular waveguide, the metasurface is configured to have either three metallic rows (Figure S14d) or four metallic rows (Figure S14e). The geometrical parameters for the meta-atom as well as circuit parameters are detailed in Table S12 and Table S13. In the measurement results, we observed a reduction in the Q factor as depicted in Figure S14f (see the broadened operating bandwidths). Consequently, the lowered Q factor resulted in increasing transient transmittance, as shown from Figure S14g to Figure S14h, where the transmittance increased from 0.09 in the slit structure to 0.36 for the three-metallic-row structure, which indicates that the performance was four times greater than that of the previous slit structure design. The slit structure measurement results are detailed in Figure S12, and a similar dual-band equivalent measurement sample having frequency combination dependency is presented in Fig. 3d. Note that the average transmittance of this structure was also approximately 0.09 only.

To better incorporate the above results into the manuscript, we have combined the circuit simulations previously shown in Supplementary Note 4 with the new experimental validation results. Now they are presented in Supplementary Note 6 (both the circuit simulation and the experimental validation). The corresponding discussion, as well as details of design parameters (tables containing circuit and geometrical parameters) has also been moved accordingly.

Additionally, we also gently remind that the time-varying and frequency-hopping mechanism of our structures comes from “*transients in electric circuits*”, namely, DC circuit mechanisms widely known to require not only imaginary part of impedance (reactance) but also the real part (resistance) leading to energy loss. This fact is clarified in Eq. (13) of Supplementary Note 2, where time constant is no longer obtained if resistance is not used. Nonetheless, as shown above Figure S14, our new design successfully enhanced the transmittance performance by properly designing Q factors, which satisfies the request from the Reviewer to improve the transmittance performance.

Furthermore, we would like to note that energy loss can be still utilized in wireless communications as, for instance, physically unclonable functions (PUFs) and internet of things (IoT) tags, which is mentioned in the second paragraph of Subsec. 3.4:

Furthermore, the concept of frequency-hopping selectivity can be used not only as spatial filters but also as smart antennas. Such an antenna design holds promising potential across various domains, including physically unclonable functions (PUFs) [R1-R3] and internet of things (IoT) tags [R4-R5]. In the context of IoT tags, the additional degree of freedom in accordance with the frequency sequence provides more identities (IDs) and thus permits the simultaneous use of more IoT tags within the same wireless network.

Finally, we emphasize that while optimizing the metasurface transmittance is important aspect of the overall design, the primary focus of our work is centred around demonstrating the novel metasurface that operates beyond linear-time-invariant (LTI) system limitations by passively changing its response depending on the incident frequency-hopping sequences. Therefore, we kindly request the reviewer's understanding and acceptance of the current results as an initial stride towards realizing such a new selectivity. We firmly believe that future endeavours hold the potential to further enhance the metasurface's transmittance, consequently expanding its range of applications such as beamforming, sensing and IoT tags.

[R1] Gao, Y., Al-Sarawi, S. F. and Abbott, D., 2020. Physical unclonable functions. *Nature Electronics*, 3, 81–91.

[R2] Yang, M., Ye, Z., Pan, H., Farhat, M., Cetin, A.E. and Chen, P.Y., 2023. Electromagnetically unclonable functions generated by non-Hermitian absorber-emitter. *Science Advances*, 9(36), p.eadg7481.

[R3] Yang, M., Zhu, L., Zhong, Q., El-Ganainy, R. and Chen, P.Y., 2023. Spectral sensitivity near exceptional points as a resource for hardware encryption. *Nature Communications*, 14(1), p.1145.

[R4] Tashiro, M., Fathnan, A.A., Sugiura, Y., Uchiyama, A. and Wakatsuchi, H., 2022. Metasurface-inspired maintenance-free Internet of things tags characterised in both frequency and time domains. *Electronics Letters* 58, 25, 937-939.

[R5] Kortuem, G., Kawsar, F., Sundramoorthy, V. and Fitton, D., 2009. Smart objects as building blocks for the internet of things. *IEEE internet computing*, 14(1), pp.44-51.

Revised Parts: We have included totally new Figure S14, Tables S12-13, and the second and third paragraphs of newly prepared Supplementary Note 6. We have also added the corresponding discussions at the seventh line from the end of the second paragraph of Subsec. 3.4 and in the third paragraph of Section 4.

Comment 3: Besides, for the proposed complex but low efficiency communication scheme, are there any advantages comparing with other metasurface-based communication methods [5]-[8]?

[5] Hodge, J. A., Mishra, K. V., & Zaghloul, A. I. (2020). Intelligent time-varying metasurface transceiver for index modulation in 6G wireless networks. *IEEE Antennas and Wireless Propagation Letters*, 19(11), 1891-1895.

[6] Zhang, L., Chen, M. Z., Tang, W., Dai, J. Y., Miao, L., Zhou, X. Y., & Cui, T. J. (2021). A wireless communication scheme based on space-and frequency-division multiplexing using digital metasurfaces. *Nature electronics*, 4(3), 218-227.

[7] Chen, M. Z., Tang, W., Dai, J. Y., Ke, J. C., Zhang, L., Zhang, C., & Cui, T. J. (2022). Accurate and broadband manipulations of harmonic amplitudes and phases to reach 256 QAM millimeter-wave wireless communications by time-domain digital coding metasurface. *National science review*, 9(1), nwab134.

[8] Zhao, J., Yang, X., Dai, J. Y., Cheng, Q., Li, X., Qi, N. H., & Cui, T. J. (2019). Programmable time-domain digital-coding metasurface for non-linear harmonic manipulation and new wireless communication systems. *National science review*, 6(2), 231-238.

Response: We sincerely value the insightful input provided by the reviewer. We believe that the above-mentioned works are important in the respective fields, addressing unique challenges by designing transceivers based on time-varying metasurfaces [5], realizing metasurfaces with space- and frequency-division multiplexing [6], exploring modulation in harmonic frequencies [8] and integrating time-varying metasurfaces into millimeter-wave wireless systems [7]. An added unique advantage of our work, in

comparison to the metasurfaces referenced by the reviewers, lies in its capacity to increase channel number by introducing an extra degree of freedom via the manipulation of frequency-hopping signals. This aspect remains unaddressed by any of the previously mentioned studies, which predominantly concentrates on exploring and employing time-varying meta-atoms to introduce innovative modulation schemes within a novel wireless framework. Furthermore, the aforementioned metasurfaces employ an active tuning method that offers even adaptive control in accordance with surrounding environment but at the same time limits the applicability due to the presence of external energy resource (e.g., a DC source) as well as precise synchronization with transmitting antennas. In contrast, our metasurface operates based on a self-tunability mechanism exploiting the variation in the incoming source signal itself.

Revised Parts: To clarify the above points, we have incorporated Refs. [5-8] at the eighth line of the third paragraph of Introduction section.

Additional changes

A minor change has been made in Figs. 4g-h, in which the x-axis values are now corrected. Initially the x-axis values were written as 0.0 – 0.8 μs , which was modified to 9.2 – 10.0 μs . Also, we improved the figure resolution of Figs. 4e-g. p_1 of Table S9 was corrected. We also clarified the input power used in the captions of Figs. 2-5. Unnecessary white lines were removed in Figures. S10, S12 and S16. In Figure S20, unit was changed from dBm to dB. Despite these changes, we emphasize that our main conclusion remains unchanged.

Reviewers' Comments:

Reviewer #2:

Remarks to the Author:

This paper mainly demonstrated a metasurface whose transmission depends on the incident frequency sequence. However, I still think this work lacks novelty and should not be published in other proper journal. In addition, some issues need to be clarified by the authors.

1). The maximum transmission amplitude is less than 0.5 even when a proper frequency sequence is adopted in the paper. The authors should give some discussion on that.

2). The authors pointed out that the transmission of the metasurface varies when different frequency sequences are adopted. To enhance the quality of the paper, a frequency-sequence design strategy for a desired transmission should be given out.

Reviewer #3:

Remarks to the Author:

The authors optimized the structure, resulting in improvement of transmittance to a certain extent as illustrated in Fig. S14 (f)-(h). However, as to the super-cell composed by 3 types of cells for 3 operative frequencies, there is a transmittance limitation for each frequency. In principle, only 1/3 of super-cell can provide full transmittance response at a corresponding frequency. Please the authors give an explanation.

As to the hopping frequency communication scheme, I do not agree on the advantage of increase channel number. Multi- (diverse-) carrier frequency technique based on metasurface also can be found in [1], [2]. In comparison with [1], [2], it seems that the only novel contribution is the larger frequency hopping step due to the proposed metasurface structure. If this is the novel contribution, please emphasize it.

[1]Wang, S. R., Dai, J. Y., Zhou, Q. Y., Ke, J. C., Cheng, Q., & Cui, T. J. (2023). Manipulations of multi-frequency waves and signals via multi-partition asynchronous space-time-coding digital metasurface. *Nature Communications*, 14(1), 5377.

[2]Fang, X., Li, M., Li, S., Ramaccia, D., Toscano, A., Bilotti, F., & Ding, D. (2023). Diverse Frequency Time Modulation for Passive False Target Spoofing: Design and Experiment. *IEEE Transactions on Microwave Theory and Techniques*.

Responses to Reviewer 2

Comment 0: *This paper mainly demonstrated a metasurface whose transmission depends on the incident frequency sequence. However, I still think this work lacks novelty and should not be published in other proper journal. In addition, some issues need to be clarified by the authors.*

Response: We appreciate the comprehensive review of the revised paper. Your feedback has been invaluable, and we have taken the necessary steps to address the issues you raised during this review round.

Comment 1: *1). The maximum transmission amplitude is less than 0.5 even when a proper frequency sequence is adopted in the paper. The authors should give some discussion on that.*

Response: In response to the reviewer's comment, we acknowledge that the maximum transmission amplitude in our current work is indeed less than 0.5. Firstly, we would like to highlight that despite this limitation, it is essential to emphasize that our primary objective in this study was to demonstrate the frequency sequence dependence, which is a significant and novel contribution in itself. The limitation in attaining maximum amplitude, therefore, does not diminish the importance of the findings which show a novel passive device responding differently to frequency-hopping signals. Particularly, as mentioned in the previous round of review, the novel ability of the metasurface opens the door to numerous potential applications in wireless communications, including for secure electromagnetic buildings or smart shielding as well as applications in smart antennas. Secondly, while the current metasurface configuration exhibits limited transmittance, our ongoing research aims to address this constraint in the subsequent phase. Notably, our preliminary analytical, numerical, and experimental investigations, as outlined in Supplementary Note 6, have shed light on potential strategies for enhancing transmittance. As seen in Figure S13 (shown below), analytical results indicate that the limitations in transmittance stem predominantly from diode-related losses, and adjustments to the quality factor of the resonance can readily improve the transmittance. Given the current setup, the incorporation of lossy diodes is imperative. Therefore, a key focus of our future research could involve developing unit cells equipped with specially designed low-loss diodes, potentially necessitating optimization at the semiconductor device level.

Figure S13: Simplified equivalent transmission line model to represent the metasurface of Fig. 3a for analysing the contribution of the Q factor. (a) Equivalent circuit model connected to the transmission line for the single slit scenario (one unit cell). C_0 and L_0 represent the capacitive and inductive components of the metallic slit structure. L_{add} and R_{add} represent the inductive and resistive components contributed by the dielectric substrate. The circuit parameters used for the diode and the series inductor and resistor (inside the diode bridge) are the same as those used in Fig. 3. (b, c) Simulated transmittance from linear network analysis for the circuit shown in (a) with varying Q factors while considering (b) zero parasitic resistance $R_{\text{add}} = 0 \Omega$ and (c) nonzero parasitic resistance $R_{\text{add}} = 3 \Omega$ (see Table S11 for variation of C_1). (d) Equivalent circuit model connected to the transmission line for the slit scenario (one super cell). (e, f) Simulated transmittance from linear network analysis for the circuit shown in (d) with varying Q factors while considering (e) zero parasitic resistance $R_{\text{add}} = 0 \Omega$, and (f) nonzero parasitic resistance $R_{\text{add}} = 3 \Omega$. (g, h) Nonlinear analysis of transient transmittance for four parallel resonance scenarios with switched frequency. Here, the input power is set to 10 dBm. f_1 , f_2 , f_3 and f_4 are 2.0, 2.5, 3.3 and 4.1 GHz, respectively. Simulated transient transmittance with varying Q factor while considering (g) zero parasitic resistance $R_{\text{add}} = 0 \Omega$ and (h) nonzero parasitic resistance $R_{\text{add}} = 3 \Omega$.

To include the above discussion, we have added the following sentences in the third paragraph of section 4.

Importantly, we also note that the transmittance limitation primarily arises from parasitic elements inherent in the embedded circuits, notably the diodes. Given the current configuration, the inclusion of these lossy diodes is imperative. Hence, prospective investigations could focus on developing unit cells equipped with specially designed low-loss diodes, potentially with an optimization from the semiconductor device level.

Revised Parts: Please refer to the last part of the third paragraph of section 4.

Comment 2: 2). *The authors pointed out that the transmission of the metasurface varies when different frequency sequences are adopted. To enhance the quality of the paper, a frequency-sequence design strategy for a desired transmission should be given out.*

Response: We appreciate the idea of providing a frequency-sequence design strategy to achieve desired transmissions with our metasurface. In our metasurface design, varying the transmission for different frequency sequences can be achieved by altering the physical connections among the unit cells working for different frequencies. As depicted in the newly introduced Figure S20a, even with the same unit cell alignment including three different operating frequencies, we can arrange connection lines to determine the frequency sequence. In this instance, simply by altering the physical connection between unit cells, we can change the correct sequence from $f_1f_2f_3$ to $f_1f_3f_2$. However, in certain designs, there may be nonnegligible coupling between unit cells, which lowers the transmittance performance. In such cases, alternatively, we can fix the connection lines but vary the capacitance values. This is exemplified in Figure S20b, where the capacitance values in unit cells 2 and 3 are swapped to change the frequency sequence from $f_1f_2f_3$ to $f_1f_3f_2$. For future research, we can further enhance the design strategy by introducing electronic switches to dynamically adjust the connections between unit cells or by incorporating varactors to tune the operating frequency. This smart and electronically controlled design approach would enable real-time adaptation of the metasurface's response to different frequency sequences, optimizing its performance for specific applications. We thank the reviewer for this valuable input, and we believe that this direction holds promise for future investigations.

To incorporate the above discussion, we have added the following paragraph in Supplementary Note 7, in which a new Figure S20 was included.

As a design strategy for the preferable sequence of frequencies, one may change connection lines among unit cells. As depicted in Figure S20a, even with the same unit cell alignment including three different operating frequencies, connection lines can be arranged to determine the frequency sequence that maximizes the transmittance. In this instance, by altering the physical connection between unit cells, the preferable sequence is changed from $f_1f_2f_3$ to $f_1f_3f_2$. However, some conducting geometry designs may show nonnegligible coupling between unit cells, which lowers the transmittance performance. In this case,

alternatively the capacitance values can be changed as shown in Figure S20b, where the capacitance values used for two unit cells are swapped to change the preferable frequency sequence from $f_1f_2f_3$ to $f_1f_3f_2$.

Figure S20. Design strategy for the preferable sequence of frequencies within a supercell configuration. (a) The same supercell with different physical connections. (b) The same supercell with different capacitance values.

Also, in the last part of Subsec. 3.3, we have added the following sentence:

Supplementary Note 7 summarizes more results related to Fig. 4 as well as design strategies for preferable sequence of frequencies within the same supercell configuration.

Revised Parts: Please see the last paragraph in Supplementary Note 7 as well as Figure S20. Also, an additional change is seen in the last sentence of Subsec. 3.3.

Responses to Reviewer 3

Comment 1: *The authors optimized the structure, resulting in improvement of transmittance to a certain extent as illustrated in Fig. S14 (f)-(h). However, as to the super-cell composed by 3 types of cells for 3 operative frequencies, there is a transmittance limitation for each frequency. In principle, only 1/3 of super-cell can provide full transmittance response at a corresponding frequency. Please the authors give an explanation.*

Response: We appreciate the reviewer's concern regarding the transmittance level of the metasurface and its relationship with the multi-resonant configuration used. In responding to this comment, it is important to note that the transmittance of each resonant unit cell is not bound by *the physical area* or the ratio of the unit cell area over the entire cross-section. Rather, in such a multi-resonant configuration, *the effective area* of the unit cell plays an important role. When we divide a supercell structure into three cells, each associated with a different resonance frequency, it does not inherently mean that the transmittance is limited to 1/3. In fact, the overall transmittance can still reach 100 % if the effective area of each cell covers the entire supercell area.

This assertion is evidenced by our simulation results of Figure R1 (shown below) in which we model a structure consisting of three slits with embedded capacitors (no diodes/nonlinear elements involved). Here, geometrical parameters for the unit cell is the same as those used in Fig. 2 in the main manuscript (see the parameters detailed in Table S2). Using this simple configuration, and by adjusting the additional capacitance values ($C_1 = 1\text{ pF}$, $C_2 = 1.5\text{ pF}$ and $C_3 = 2\text{ pF}$), we see that the transmittance can be maximized at 1.67 GHz, 1.97 GHz and 2.39 GHz, all with near unity amplitude. The structure is in deep subwavelength dimensions, indicating a sufficiently large effective area of each unit cell which well spans the entire supercell. We can also see from the field profile that high intensity fields are observed at different slits corresponding to different resonance frequencies. This indicates that each unit cell plays its distinct role in creating maximum transmittance. Similar results have also been reported in other works including Refs [a] and [b]. Particularly, in Ref [a], symmetrical three-slot structure is used as a supercell which results in a high gain dual-band transmitarray.

Figure R1. Simulation of a metasurface supercell consisting of three slits with embedded capacitors (no diodes/nonlinear elements involved). (a) Metasurface supercell model. (b) Transmission and reflection amplitude. (c) Electric field profiles at three corresponding resonance frequencies.

- [a] Wu, R.Y., Li, Y.B., Wu, W., Shi, C.B. and Cui, T.J., 2017. High-gain dual-band transmitarray. IEEE Transactions on antennas and Propagation, 65(7), pp.3481-3488.
- [b] Huang, S., Xie, Z., Chen, W., Lei, J., Wang, F., Liu, K. and Li, L., 2018. Metasurface with multi-sized structure for multi-band coherent perfect absorption. Optics express, 26(6), pp.7066-7078.

Regarding the result in Figures S14 (f)-(h) which show non-maximized transmittances, we clarify that this is due to the presence of parasitic losses inherent within the diodes, which has also been confirmed in the equivalent circuit analysis shown in Figure S13. Nevertheless, we have demonstrated in the same figure that by modifying the Q factor as in the case of 3-metallic-bar configuration, the transmittance can still be improved.

In Subsection 3.1 of the manuscript, we have provided discussion on multiband operation of the metasurface. Therefore, to include the above discussion, we have added the following sentences in the first paragraph of subsection 3.1, further clarifying the multiband design of the metasurface.

Note that despite the limited physical area of the unit cells, the transmittance at these two frequencies was maximized due to the large effective area of the resonant cells covering the entire supercell. This observation aligned with findings from other reported works where efficient multiband operation arose from the combination of different subwavelength elements with distinct resonance frequencies [a, b].

Revised Parts: Please refer to the first paragraph of subsection 3.1 and the list of references.

Comment 2: *As to the hopping frequency communication scheme, I do not agree on the advantage of increase channel number. Multi- (diverse-) carrier frequency technique based on metasurface also can be found in [1], [2]. In comparison with [1], [2], it seems that the only novel contribution is the larger frequency hopping step due to the proposed metasurface structure. If this is the novel contribution, please emphasize it.*

[1] Wang, S. R., Dai, J. Y., Zhou, Q. Y., Ke, J. C., Cheng, Q., & Cui, T. J. (2023). Manipulations of multi-frequency waves and signals via multi-partition asynchronous space-time-coding digital metasurface. Nature Communications, 14(1), 5377.

[2] Fang, X., Li, M., Li, S., Ramaccia, D., Toscano, A., Bilotti, F., & Ding, D. (2023). Diverse Frequency Time Modulation for Passive False Target Spoofing: Design and Experiment. IEEE Transactions on Microwave Theory and Techniques.

Response: We appreciate the reviewer's perspective on the metasurface and its relation to channel number in communication scheme. The previous works in references [1] and [2] demonstrate an increase in channel number by modulating the frequency of an incident wave using diode-based metasurfaces through electrical biasing. In [1], the multi-frequency operation was used for multiplexing data over single metasurface reflector while in [2] it was used as an efficient radar false target spoofing as seen in Table R1 below. The frequency modulation relies on the clocking speed of an electronic system (FPGA), leading to fine incremental steps in the modulated frequency. In addition to this distinct operation, it is crucial to distinguish our metasurface approach from those presented in [1] and [2]. In our work, the metasurface does not actively modulate the incident wave from an external biasing circuit, and it is not designed to generate harmonics or multi-frequency waves. Instead, our metasurface functions passively by filtering the incident wave based on the frequency hopping signal it encounters. The larger step size in the frequency hopping observed in our work is indeed a result of the multiband nature of the unit cells employed and not by external biasing of the unit cell.

Table R1. Comparison between metasurfaces with diverse carrier frequency and our work.

Title	Novel contribution	Method
[1] Manipulations of multi-frequency waves and signals via multi-partition asynchronous space-time-coding digital metasurface	A strategy to enhance number of independently controlled high-order harmonics in diode-based metasurface with simplified coding algorithms	Simulation, experiment
[2] Diverse Frequency Time Modulation for Passive False Target Spoofing: Design and Experiment	Deceptive technique for generating false targets in radar detection using multi-frequency harmonics produced by time-modulated metasurfaces.	Simulation, experiment
Our work	A passive metasurface filtering incident multi-frequency signals based on the specific sequence of frequency hopping.	Simulation, experiment

Regarding the expansion of channel capacity, the approach employed in [1] and [2] does result in a higher number of frequency carriers, contributing to improved channel capacity. In these metasurfaces, by assigning frequency resources based on a linear frequency concept, the number of channels is maximized by increasing the frequency number N (channel number = N). For example, in Ref [1], the channel number generated by the metasurface may be increased to 8, which is the same as the number of generated frequency carrier. In contrast, our paper introduces a concept where the channel number can increase through diverse sequences of frequency hopping, following a factorial of the frequency number (channel number = $N!$). Consequently, thanks to the innovative filtering method facilitated by our metasurface, we can overcome the limitations of a linear frequency concept, greatly enhancing available channels. We trust that this clarification illuminates the distinctive contribution of our metasurface, which has the potential to further drive new research in the related area.

To include the above discussion, we have added the following sentence in the first paragraph of section 4.

Also, time-varying metasurfaces have been utilized to increase the number of carrier frequencies through harmonics generation, while the number of channels was still linearly proportional to the frequencies generated by signal sources and modulated metasurfaces [1,2].

Revised Parts: Please refer to the first paragraph of section 4 and the list of references.

Reviewers' Comments:

Reviewer #2:

Remarks to the Author:

I think the paper is well revised and it is suitable for publication now as it is.

Reviewer #3:

Remarks to the Author:

The authors have addressed all my comments clearly.

Responses to Reviewer 2

Comment 1: *I think the paper is well revised and it is suitable for publication now as it is.*

Response: We thank the reviewer for recommending our paper for publication.

Responses to Reviewer 3

Comment 1: *The authors have addressed all my comments clearly.*

Response: We thank the reviewer for recommending our paper for publication.